# Antibody Fc-receptor FcεR1γ stabilizes cell surface receptors in group 3 innate lymphoid cells and promotes anti-infection immunity

Chao Huang [1,2,3,4,7] ✉, Wenting Zhu[2,3,4,5,7], Qing Li [6,7], Yuchen Lei[1,7],
Xi Chen[2,3,4], Shaorui Liu[2,3,4], Dianyu Chen[2,3,4], Lijian Zhong[1], Feng Gao[2,3,4],
Shujie Fu[2,3], Danyang He[2,3], Jinsong Li [1,6] & Heping Xu [2,3,4] ✉

Group 3 innate lymphoid cells (ILC3) are crucial for maintaining mucosal homeostasis and regulating inflammatory diseases, but the molecular mechanisms governing their phenotype and function are not fully understood. Here, we show that ILC3s highly express *Fcer1g* gene, which encodes the antibody Fc-receptor common gamma chain, FcεR1γ. Genetic perturbation of FcεR1γ leads to the absence of critical cell membrane receptors NKp46 and CD16 in ILC3s. Alanine scanning mutagenesis identifies two residues in FcεR1γ that stabilize its binding partners. FcεR1γ expression in ILC3s is essential for effective protective immunity against bacterial and fungal infections. Mechanistically, FcεR1γ influences the transcriptional state and proinflammatory cytokine production of ILC3s, relying on the CD16-FcεR1γ signaling pathway. In summary, our findings highlight the significance of FcεR1γ as an adapter protein that stabilizes cell membrane partners in ILC3s and promotes anti-infection immunity.

Innate lymphoid cells (ILC) are a diverse group of tissue-resident effector lymphocytes that play essential roles in programming mucosal homeostasis and inflammation[1,2]. Among ILC subsets, group 3 ILCs (ILC3) have been identified as particularly crucial for intestinal immunity[2–4]. ILC3s consist of distinct populations, including CCR6+ lymphoid tissue inducer (LTi) cells, NCR+ (NCR: NKp46 in mice, NKp44 in humans) cells, and CCR6−NCR− (double negative, DN) cells. LTi cells support the formation of gut-associated lymphoid tissues[5], while DN ILC3s and NCR+ILC3s emerge postnatally in response to microbial stimulation[6–8]. The functions of ILC3s are mediated through the secretion of various cytokines, including IL-22, IL-17A, IFN-γ, and GM-CSF, as well as MHCII-mediated antigen presentation[9]. Dysregulation

of ILC3-mediated immune tolerance to commensal microbiota has been associated with inflammatory bowel disease (IBD)[10,11], whereas compromised ILC3 effector function can impair host defense against pathogens like *Citrobacter rodentium* (*C. rodentium*)[9,12]. Therefore, tight regulation of ILC3 activity is crucial for maintaining intestinal immune homeostasis.

ILC3 activation is independent of receptor-mediated antigen signaling due to the lack of antigen-specific receptors[4,13]. Instead, their responses are programmed by cell surface receptors that recognize various environmental signals, including cytokine receptors, metabolite receptors, and nervous system signal receptors[14]. However, the receptors that dictate the phenotype and function of ILC3s are not yet

[1]Key Laboratory of Systems Health Science of Zhejiang Province, School of Life Science, Hangzhou Institute for Advanced Study, University of Chinese Academy of Sciences, Hangzhou, China. [2]Laboratory of Systems Immunology, School of Medicine, Westlake University, Hangzhou, Zhejiang, China. [3]Key Laboratory of Growth Regulation and Translational Research of Zhejiang Province, School of Life Sciences, Westlake University, Hangzhou, Zhejiang, China. [4]Westlake Laboratory of Life Sciences and Biomedicine, Hangzhou, Zhejiang, China. [5]School of Pharmaceutical Science and Technology, Hangzhou Institute for Advanced Study, University of Chinese Academy of Sciences, Hangzhou, China. [6]Key Laboratory of Multi-Cell Systems, Shanghai Key Laboratory of Molecular Andrology, CAS Center for Excellence in Molecular Cell Science, Shanghai Institute of Biochemistry and Cell Biology, Chinese Academy of Sciences, University of Chinese Academy of Sciences, Shanghai, China. [7]These authors contributed equally: Chao Huang, Wenting Zhu, Qing Li, Yuchen Lei. ✉e-mail: huangchao@ucas.ac.cn; xuheping@westlake.edu.cn

fully understood, and the molecular mechanisms regulating the abundance of cell membrane receptors in ILC3s remain to be studied.

FcεR1γ, which is encoded by the *Fcer1g* gene, serves as a shared subunit of multiple Fc receptors (FcRs)[15]. FcεR1γ does not directly bind any extracellular ligand. Instead, it acts as an adapter protein that facilitates downstream signaling when its membrane partners recognize and bind their extracellular ligands. For example, *Fcer1a* expressed on mast cells binds to IgE, while *Fcgr1*, *Fcgr3*, *Fcgr4* (*FCGR1A* and *FCGR3A* in human) expressed on myeloid and NK cells binds to IgG[15]. Upon ligand binding, FcεR1γ activates the SYK-RAS-MAPK pathway via an immunoreceptor tyrosine-based activation motif (ITAM) present in its cytoplasmic tail. Additionally, FcεR1γ is involved in myeloid cell activation in response to the recognition of pathogen-associated molecular patterns (PAMPs) by C-type lectin receptors (CLRs)[16]. Moreover, FcεR1γ has been reported as a component of the IL-3 receptor in basophils[17]. However, the expression and function of FcεR1γ and its binding receptors in ILC3s are still unclear and warrant further investigation.

Here by analyzing the expression of different FcRs in intestinal ILCs, we find that ILC3s highly express *Fcer1g* and *Fcgr3*, the latter encoding CD16 (also known as FcγRIII). FcεR1γ is dispensable for ILC3 development and their transcriptional states. However, it maintains the protein abundance of CD16 and the major phenotypic molecule NKp46, both of which are known binding partners of FcεR1γ. We show that two essential residues in FcεR1γ are required to stabilize its binding proteins across different cell types. FcεR1γ expression in ILC3s promotes the host defense against local *C. rodentium* and systemic *Candida albicans* (*C. albicans*) infections by promoting IL-17A and IL-22 secretion. Importantly, CD16, but not NKp46, plays crucial roles in ILC3 functions, likely through the FcεR1γ-mediated immune activation signaling pathway. In conclusion, our findings reveal that FcεR1γ functions as a post-transcriptional regulator, maintaining the abundance of cell surface receptors in ILC3s necessary for effective anti-infection immunity. This study provides insights into the molecular mechanisms governing ILC3 function and highlights the importance of FcεR1γ in regulating ILC3-mediated intestinal immunity.

## Results

### FcεR1γ maintains the expression of major phenotypic proteins in ILC3s

Through analysis of the published single-cell atlas of immune cells in the small intestine of mice (Supplementary Fig. S1A, B)[18], we found that both LTi cells, also known as CCR6⁺ ILC3s, and CCR6⁻ ILC3s exhibited a high expression of *Fcer1g* gene (Fig. 1A). Consistently, the expression of *FCER1G* in human intestinal ILC3s was also observed[19] (Supplementary Fig. S1C, D), indicating a conserved expression pattern of this gene in ILC3s. Additionally, ILC3s expressed the *Fcgr3* gene, which encodes CD16, while the expression of other FcR genes was not detected (Fig. 1A). Furthermore, we confirmed the co-expression of FcεR1γ and CD16 at the protein level in ILC3s isolated from the intestines of mice (Fig. 1B).

To investigate the cell-intrinsic function of FcεR1γ in ILC3s, we generated *Fcer1g*-conditional knockout (*Rorc-cre⁺Fcer1g^f/f*, CKO) mice by crossing *Fcer1g^flox/flox* mice with *Rorc-cre* mice (Supplementary Fig. S1E–G), which are widely used for targeting ILC3s[20,21]. The specific deletion of FcεR1γ expression in RORγt-expressing cells was confirmed in CKO mice compared to *Rorc-cre⁺Fcer1g^+/+* (control) mice (Fig. 1C). Importantly, CD3⁺ T cells, including RORγt⁺ Th17 cells, Treg cells and γδT cells, which are also targeted by the *Rorc-cre*[22], did not co-express FcεR1γ with RORγt (Fig. 1C). Thus, using this CKO mouse model, we were able to determine the cell-intrinsic function of FcεR1γ in ILC3s.

We observed comparable frequencies and numbers of both CCR6⁺ and CCR6⁻ ILC3s in the small intestine and colon between control and CKO mice at the steady state (Fig. 1D, E, and Supplementary Fig. S1H). The production of IL-17A and IL-22 in ILC3s was also

comparable between control and CKO mice (Supplementary Fig. S1I). However, the deficiency of FcεR1γ in ILC3s significantly abolished the expression of NKp46, a major phenotypic protein of CCR6⁻ ILC3s[23] (Fig. 1F). In addition, the expression of CD16, another binding partner of FcεR1γ, was also significantly reduced in both FcεR1γ-deficient CCR6⁺ and CCR6⁻ ILC3s compared to the control cells (Fig. 1G and Supplementary Fig. S1J).

To further determine whether FcεR1γ influenced the transcriptional states and proportions of different ILC3 subtypes, we analyzed intestinal ILCs of CKO and control mice via scRNA-seq (Fig. 1H, and Supplementary Fig. S1K–N). Computational analyses revealed that the absence of FcεR1γ did not significantly alter the proportions (Fig. 1I) and transcriptional states (Supplementary Fig. S1M) of different ILC3-subtypes. Importantly, *Ncr1* and *Fcgr3* transcript abundance in ILC3s remained intact in the CKO mice compared to control mice (Fig. 1J, K, and Supplementary Fig. S1N). These findings demonstrated that FcεR1γ maintains the expression of its binding partners, including NKp46 and CD16, in ILC3s through a post-transcriptional regulatory mechanism.

### Two essential residues in FcεR1γ for maintaining its binding partners

We noticed the absence of CD64, encoded by the *Fcgr1* gene[24], in FcεR1γ-deficient bone marrow-derived macrophages (BMDMs) (Supplementary Fig. S2A). This suggests a conserved mechanism by which FcεR1γ maintains its binding partners across different cell types. To investigate the specific amino acid residues in FcεR1γ that are essential for maintaining the abundance of its binding partners, we performed alanine scanning mutagenesis analysis in BMDMs (Fig. 2A), which are more amenable to culture and transfection in vitro compared to ILC3s. *Fcer1g⁻/⁻* BMDMs were infected with lentiviruses expressing either WT FcεR1γ or variants with single mutations in the extracellular and transmembrane domains, based on the predicated importance of specific amino acids for binding partner interactions. The expression of WT FcεR1γ robustly increased the membrane abundance of CD64 (Fig. 2B, C, and Supplementary Fig. S2B). However, the expression of FcεR1γ carrying D29A or L39A mutations failed to effectively restore CD64 expression, indicating that D29 and L39 are essential for supporting CD64 expression in BMDMs.

To determine the function of D29 and L39 residues in FcεR1γ in maintaining NKp46 and CD16 expression in ILC3s, we isolated intestinal ILC3s for in vitro expansion and transfection with lentivirus expressing WT FcεR1γ, FcεR1γ^D29A or FcεR1γ^L39A (Supplementary Fig. S2C–E, **Methods**). Although the overall transfection efficiency in ILC3s was low (Supplementary Fig. S2F), the expression of exogenous WT FcεR1γ enabled the detection of a subset of NKp46⁺ cells in CCR6⁻ ILC3s (Fig. 2D). However, the expression of FcεR1γ^D29A or FcεR1γ^L39A failed to induce the expression of NKp46 (Fig. 2D, E). Notably, the expression of WT FcεR1γ did not induce NKp46 expression in CCR6⁺ ILC3s, which are known to lack *Ncr1* gene transcripts. Additionally, we found that WT FcεR1γ, but not the FcεR1γ^D29A or FcεR1γ^L39A, increased the expression of CD16 in both CCR6⁻ and CCR6⁺ ILC3s (Supplementary Fig. S2G, H). These findings demonstrated that the aspartic acid at position 29 and leucine at position 39 are essential and conserved amino acids in FcεR1γ for maintaining the abundance of its binding partners, including NKp46 and CD16 in ILC3s, as well as CD64 in macrophages.

### FcεR1γ promoted ILC3-mediated protective immunity against bacterial infection

To investigate the function of FcεR1γ in ILC3s in inflammatory conditions, we employed dextran sulfate sodium (DSS)-induced colitis model in control and CKO mice (Supplementary Fig. S3A). The deficiency of FcεR1γ in ILC3s slightly alleviated the weight loss caused by DSS (Supplementary Fig. S3B). However, there were no significant

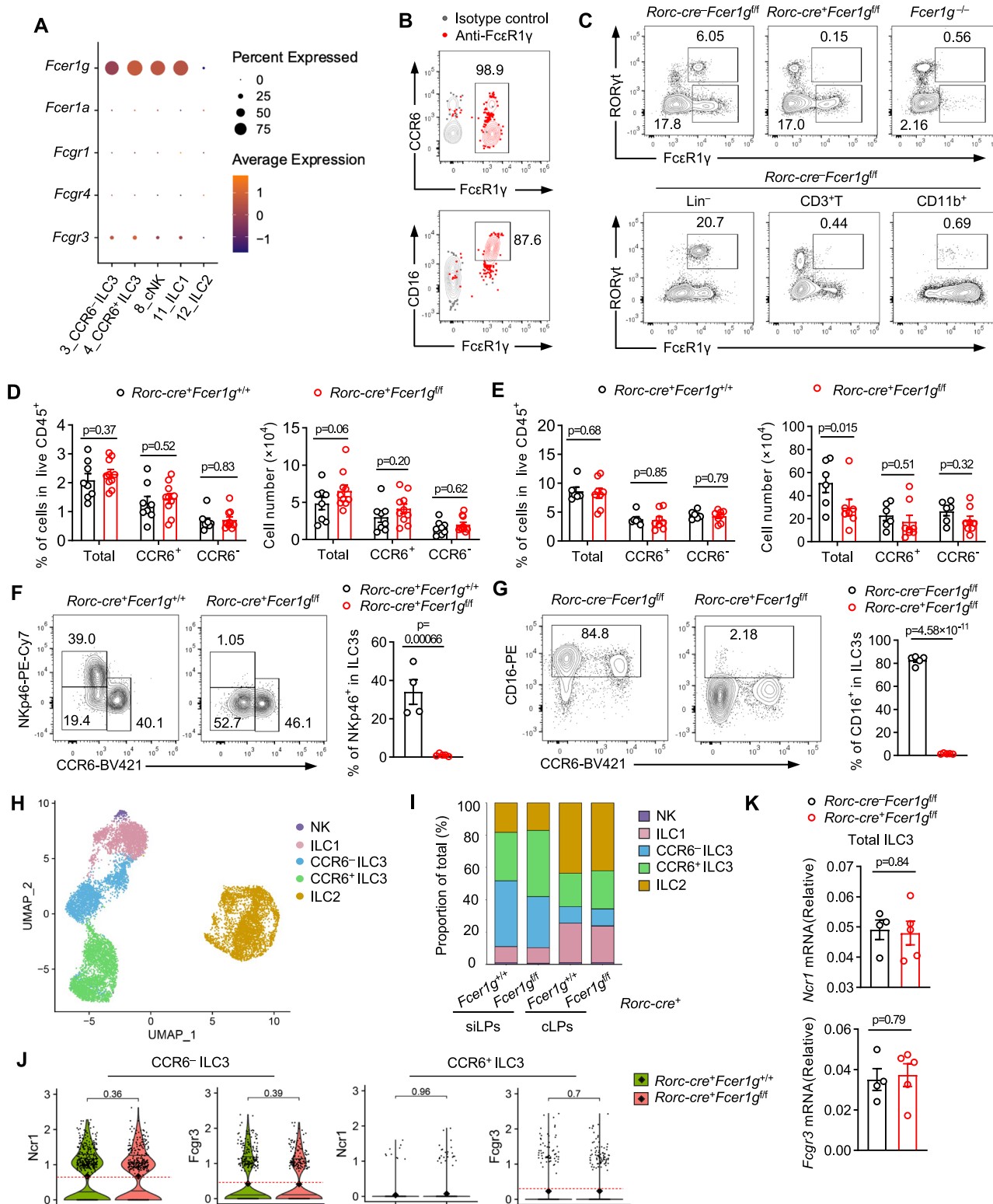

differences observed in colon length (Supplementary Fig. S3C), intestinal histology score (Supplementary Fig. S3D), frequencies of total ILC3s (Supplementary Fig. S3E) or IL-22 or IL-17A-expressing ILC3s (Supplementary Fig. S3F, G) between CKO and control mice. These results suggested that FcεR1γ may not play a critical role in ILC3s in response to DSS-induced inflammation.

ILC3s have been reported to contribute to host defense against pathogens, such as intestinal *C. rodentium*, mainly through the production of IL-22[12]. To determine whether FcεR1γ was involved in the

ILC3-mediated anti-infection immunity, we used a bioluminescent strain (ICC180) of *C. rodentium*[25] to trigger an immune response in the intestine (Fig. 3A). We found that loss of FcεR1γ in ILC3s significantly aggravated tissue damage caused by the *C. rodentium* infection[26] (Fig. 3B, C). Although the weight loss was slightly more severe in CKO mice compared to control mice after infection (Supplementary Fig. S3H), it was relatively mild in both groups, likely due to the low toxicity of ICC180 strain[25]. Nonetheless, the bacterial loads in the stool, intestine, and living animals of CKO mice were significantly increased

**Fig. 1 | FcεR1γ is dispensable for ILC3 development but specifically required for the expression of major phenotypic proteins. A** Percentages (dot size) and mean expression (color, Z score of average log₂(TPM + 1)) of genes encoding antibody Fc receptors in different ILC clusters in Supplementary Fig. S1A. **B** Representative flow plots showing the expression of FcεR1γ (top) and its co-expression with CD16 (bottom) in ILC3s from small intestinal lamina propria (siLP) of WT mice. **C** Representative flow plots showing the expression of FcεR1γ in RORγt⁺ and RORγt⁻ cell compartments in CD45⁺ cells from siLPs with the indicated genotype (top). Representative flow plots showing expression of FcεR1γ and RORγt in Lin⁻ (CD45⁺CD19⁻CD3⁻CD11b⁻), CD3⁺ T and CD11b⁺ cells in *Rorc-cre⁻Fcer1g^{f/f}* mice (bottom). **D, E** The percentages and numbers of total, CCR6⁺ and CCR6⁻ ILC3 compartments in colonic lamina propria (cLP) (**D**, *Rorc-cre⁺Fcer1g^{+/+}*, n = 8 mice; *Rorc-cre⁺Fcer1g^{f/f}*, n = 10 mice) and siLPs (**E**, *Rorc-cre⁺Fcer1g^{+/+}*, n = 6 mice; *Rorc-cre⁺Fcer1g^{f/f}*, n = 8 mice) as gated in Supplementary Fig. S1H at the steady state. **F** Representative flow plots showing the expression of NKp46 and CCR6 in ILC3s from siLPs at the

steady state (left) and the frequencies of NKp46⁺ cells (right, *Rorc-cre⁺Fcer1g^{+/+}*, n = 4 mice; *Rorc-cre⁺Fcer1g^{f/f}*, n = 5 mice). **G** Representative flow plots showing CD16 expression in ILC3s from siLPs at the steady state (left) and the frequencies of CD16⁺ cells (right, n = 5 mice per group). **H-J** scRNA-seq analyzes of ILCs in siLPs and cLPs of control and CKO mice. Uniform manifold approximation and projection (UMAP) plot of intestinal ILCs (dots), colored by cluster membership (**H**). Bar plots showing the proportions of each ILC subset with the indicated genotype (**I**). Violin plot of the distribution of mean expression (log₂(TPM + 1), y axis) of *Ncr1* and *Fcgr3* in CCR6⁺ and CCR6⁻ ILC3s from siLPs (**J**). **K** Quantification of the abundance of *Ncr1* and *Fcgr3* transcripts relative to *Gapdh* in ILC3s sorted from siLPs of mice (as gated in Fig. S1F, *Rorc-cre⁻Fcer1g^{f/f}*, n = 4 mice; *Rorc-cre⁺Fcer1g^{f/f}*, n = 5 mice). Data are representative of three (**B, C, F, G**) or are pooled from three (**D, E**), or two (**K**) independent experiments shown as the mean ± SEM. Statistical significance was tested by two-tailed *t* test (**D, E, F, G, K**) or two-tailed wilcoxon test (**J**).

compared to the control mice (Fig. 3D, E, and Supplementary Fig. S3I–K). Consistent with previous studies that demonstrated the importance of cytokine production, especially IL-22 and IL-17A, from intestinal ILC3s in defense against *C. rodentium*[12], we found that the frequencies of both IL-22- and IL-17A-expressing ILC3s were significantly decreased in CKO mice compared to control mice (Fig. 3F–I, and Supplementary Fig. S3L). Thus, FcεR1γ supported ILC3-mediated host defense immunity against bacterial infection.

## FcεR1γ expression in ILC3s enhanced host immune defense against invasive fungal infection

Given that FcεR1γ expression in ILC3s participated in the inflammation induced by the bacterial infection but not DSS treatment, we further investigated FcεR1γ function in ILC3s in response to other pathogens. Previous studies have shown that IL-17 secreted from ILCs plays an important role in the sublingual infection of *C. albicans*[27,28], which is normally a harmless commensal organism but can be an opportunistic pathogen for immunologically weak and immunocompromised people[29]. *C. albicans* is unable to colonize and infect the intestines of wild-type mice unless pre-treated with antibiotics or DSS[30,31], both of which perturb ILC3 development and activation. Therefore, we employed a systemic infection model through intravenous injection of *C. albicans*[32] to study the response of ILC3s (Fig. 4A). Moreover, systemic *C. albicans* infection induced severe mortality, which allows us to directly determine the protective immunity conferred by FcεR1γ expression in ILC3s. We observed that the infection significantly induced the expression of *Il22* and *Il17a* in ILC3s, particularly in the CCR6⁺ cell compartments (Supplementary Fig. S4A, B). Bulk RNA-seq analysis further revealed that ILC3s underwent dramatic transcriptional changes in response to fungal infection (Supplementary Fig. S4C–E). Notably, the genes induced by the infection were enriched in multiple immune activation pathways, including cytokine-cytokine receptor interaction (e.g., *Il22, Tnfrsf9 and Clcf1*), JAK-STAT signaling pathway (e.g., *Stat3*), and chemokine signaling pathway (e.g., *Ccr7*) (Supplementary Fig. S4F, G)[33–35]. These findings demonstrated that intestinal ILC3s actively responded to the systemic *C. albicans* infection.

Importantly, the deficiency of FcεR1γ in ILC3s significantly accelerated the fatality induced by the fungal infection (Fig. 4B). Additionally, CKO mice manifested increased fungal load in the kidneys (Fig. 4C). Consistently, the kidneys of CKO mice showed exacerbated fungal hyphae, inflammatory cell infiltration, and tissue damage compared to control mice (Fig. 4D–F). These observations demonstrated that the deficiency of FcεR1γ in ILC3s sensitized the host to systemic fungal infection.

## FcεR1γ programmed the transcriptional states and effector function of ILC3s for antifungal immunity

To investigate the underlying mechanisms by which FcεR1γ is involved in ILC3-mediated antifungal immunity, we analyzed the transcriptome

of ILC3s of CKO and control mice after fungal infection. The deficiency of FcεR1γ substantially rewired the transcriptional state of intestinal ILC3s in response to the infection (Supplementary Fig. S5A–E). Gene set enrichment analysis (GSEA) revealed that FcεR1γ-deficient ILC3s exhibited reduced expression of genes enriched in multiple immune activation pathways, including cytokine-cytokine receptor interaction, JAK-STAT signaling and chemokine signaling pathways (Supplementary Fig. S5F), all of which were induced by the infection in WT ILC3s (Supplementary Fig. S4F, G). Notably, FcεR1γ-deficient ILC3s showed a significant decrease in the expression of infection-induced immune activation genes, such as *Fcgr3, Ccr7, Il22, Il17a* and *Cd93* (Supplementary Fig. S5G). Direct quantification of the expression of infection-induced genes confirmed that their overall expression was significantly diminished in FcεR1γ-deficient ILC3s compared to WT cells (Fig. 5A, B).

A recent study has suggested that a subset of ILC3s in lymph nodes can sense *C. albicans* via receptors such as TLR2, Dectin-1, CD80, and CD86, and subsequently present fungal antigens through MHC-II to promote host defense against fungal infection[36]. We found that the deficiency of FcεR1γ did not influence the expression of these receptors in ILC3s (Supplementary Fig. S6A, B). To further assess whether FcεR1γ deficiency impaired ILC3 effector function in response to fungal infection, cytokine production of ILC3s in CKO and control mice was measured. While the frequencies of total ILC3s remained unchanged in the CKO mice compared to controls (Fig. 5C), the deficiency of FcεR1γ significantly compromised the frequencies of IL-17A- and IL-22-secreting ILC3s (Fig. 5D–G). Notably, the production of IL-17A and IL-22 in Lin⁺ cell compartment was comparable between CKO and control mice (Supplementary Fig. S6C), indicating a specific defect in ILC3s. We also confirmed that intestinal αβT or γδT cells, which are targeted by the *Rorc-cre*, still did not express FcεR1γ under the infection condition (Supplementary Fig. S6D, E). These findings collectively demonstrated that FcεR1γ-mediated signaling influenced the transcriptional state and effector function of ILC3s in response to fungal infection.

We then investigated the binding partners of FcεR1γ to further explore its mediated signaling pathways. Given the conserved functions of FcεR1γ in ILC3s and macrophages (Fig. 2), we utilized macrophage cell line RAW264.7 cells for biochemical assays. After exogenously expressing FcεR1γ-FLAG proteins, we performed immunoprecipitation-mass spectrometry (IP-MS) assay using FLAG antibodies to analyze the cell lysate (Supplementary Fig. S7A). The MS analysis identified dozens of proteins binding with FcεR1γ, such as ZW10, XPOT, JAK1, IPO11, NUP205 (Supplementary Fig. S7B–E). GO (Gene Ontology) analysis revealed that FcεR1γ-binding proteins were enriched in multiple biological processes and molecular functions, including nucleocytoplasmic transport and protein import into the nucleus (IPO7, TNPO1, XPO5, XPOT, IPO4, KPNB1, NUP205, XPO1, IPO9), small GTPase mediated signal transduction (Supplementary

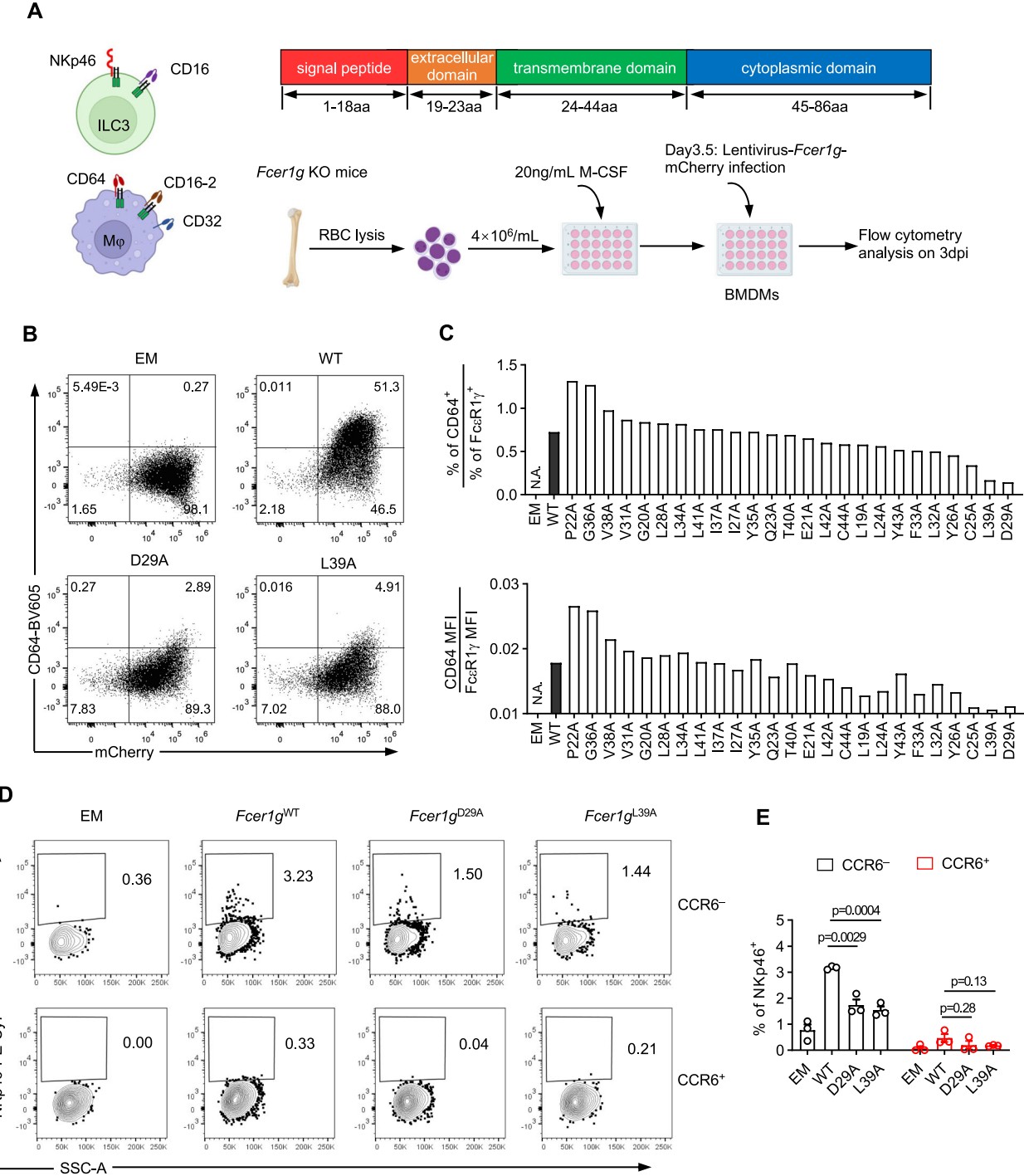

**Fig. 2 | FcεR1γ stabilizes NKp46 expression depending on aspartic acid at position 29 and leucine at position 39. A** Experimental scheme of alanine-scanning mutagenesis assay in *Fcer1g* KO BMDMs (**Methods**). BMDMs were differentiated from the bone marrow cells of *Fcer1g* KO mice, and lentivirally introduced with WT or mutated FcεR1γ for measuring CD64 expression by flow cytometry. **B** Representative flow plots showing the expression of CD64 in BMDMs infected by lentivirus that express WT, D29A or L39A FcεR1γ or empty (EM) as negative control. The infected cells were identified by mCherry expression. **C** The percentages (top) and MFIs (bottom) of CD64 relative to lentivirally expressed WT FcεR1γ or the ones carrying different single point mutations in *Fcer1g* KO BMDMs were shown. **D, E** Representative flow plots showing the expression of NKp46 in cultured ILC3s (as gated in Supplementary Fig. S2C) infected by lentivirus that express WT, D29A or L39A FcεR1γ or EM as negative control (**D**). The infected cells were identified by GFP expression (as gated in Fig. S2F). **E** quantification of the frequencies of NKp46+ in CCR6+ and CCR6− ILC3s in (**D**). $n = 3$ replicates per group. Data are representative of two (**B, C**) or are pooled from two (**D, E**) independent experiments shown as the mean ± SEM. Statistical significance was tested by one-way ANOVA followed by Tukey's multiple comparisons test (**E**).

Fig. S7F–H). Of note, the binding of JAK1 with FcεR1γ, along with PPP2R1B, PPP2R5A that belong to IL-6 signaling pathway[37] and ARFGEF1, KPNB1, ARFGEF3, GBF1, CYFIP1, ELMO1, DOCK2, HACD3 that belong to small GTPase signaling pathway[38], suggested that FcεR1γ is involved in activation cascade transduction from cytoplasmic to nucleus. Consistently, we found that the loss of FcεR1γ significantly impaired the phosphorylation of JAK1 and JAK3 in ILC3s in response to the fungal infection (Fig. 5H, I). Taken together, these data suggested

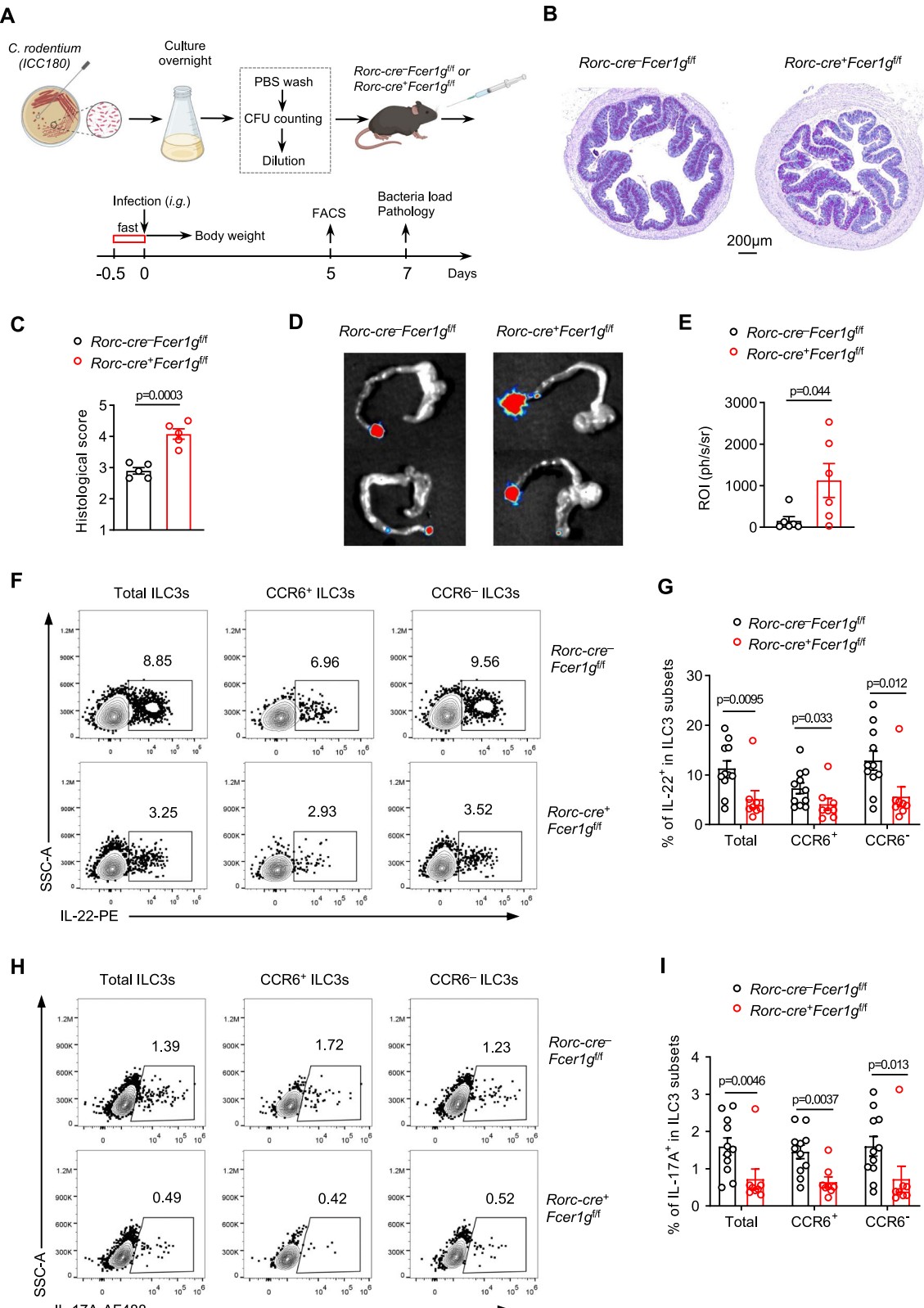

**Fig. 3 | FcεR1γ expression in ILC3s promotes host elimination of *Citrobacter rodentium*. A** Experimental scheme of *C. rodentium* infection model (**Methods**). **B** Representative periodic acid–Schiff (PAS) staining images of colons of *Rorc-cre⁻Fcer1g^f/f* and *Rorc-cre⁺Fcer1g^f/f* mice on day 7 post the infection as in (**A**). **C** Quantification of the histological scores of images as in (**B**) (*n* = 5 mice per group). **D, E** Representative images of bacterial load determined by bioluminescent imaging in the colon (**D**) and the quantification (**E**) (*n* = 6 mice per group). **F-I**

Representative flow plots showing the intracellular abundance of IL-22 (**F**) or IL-17A (**H**) in total, CCR6⁺ and CCR6⁻ ILC3s from siLPs. Quantification of the frequencies of IL-22 (**G**, *Rorc-cre⁻Fcer1g^f/f*, *n* = 11 mice; *Rorc-cre⁺Fcer1g^f/f*, *n* = 8 mice) or IL-17A (**I**, *Rorc-cre⁻Fcer1g^f/f*, *n* = 11 mice; *Rorc-cre⁺Fcer1g^f/f*, *n* = 8 mice) -expressing cells in each compartment. Data are representative of two (**B-E, F, H**) or are pooled from two (**G, I**) independent experiments shown as the mean ± SEM. Statistical significance was tested by two-tailed *t* test (**C, E, G, I**).

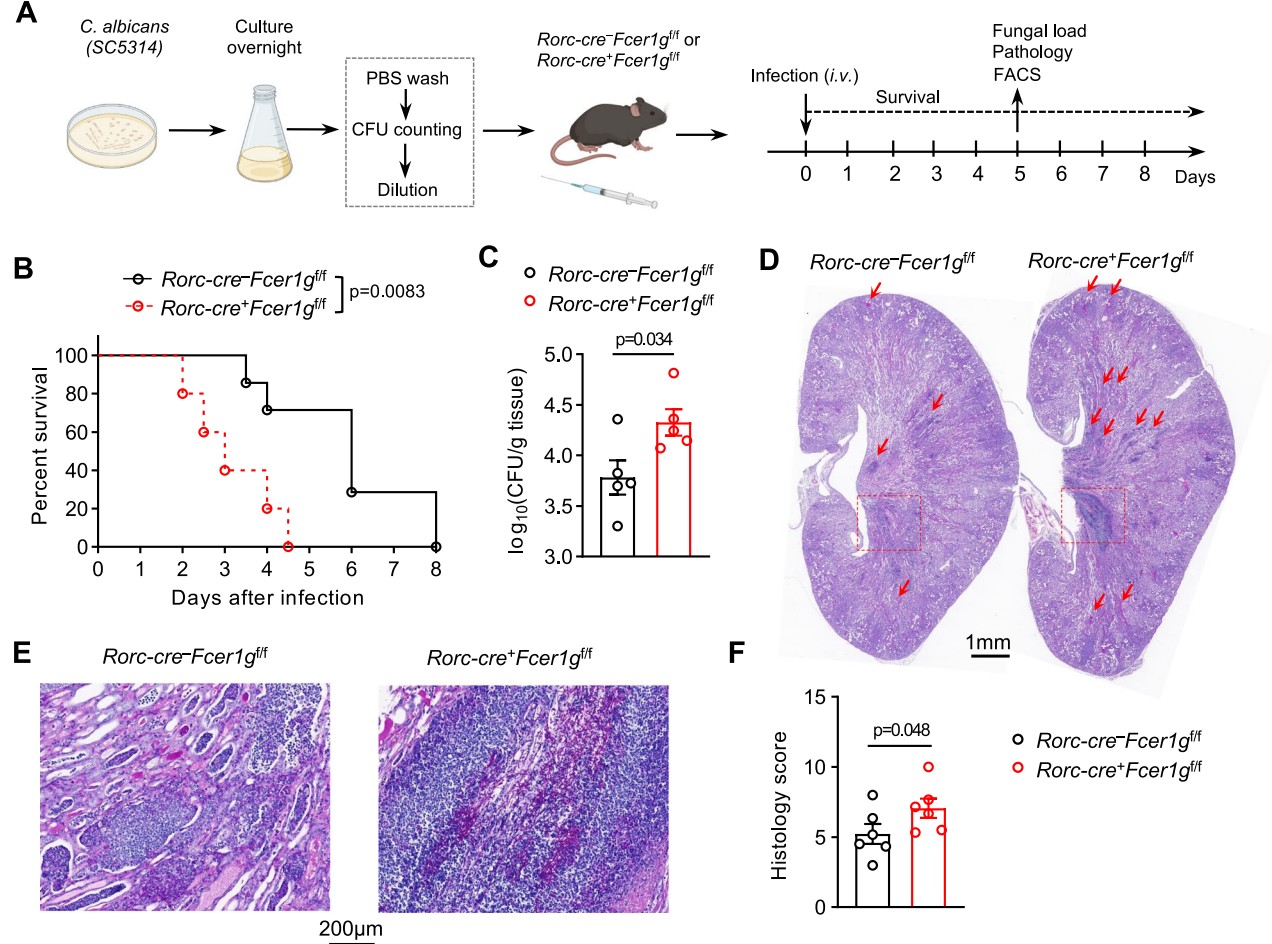

**Fig. 4 | Genetic perturbation of *Fcer1g* in ILC3s aggravates invasive fungal infection. A** Experimental scheme of *C. albicans* infection model (**Methods**). **B** Survival curves of *Rorc-cre⁻Fcer1g^{f/f}* and *Rorc-cre⁺Fcer1g^{f/f}* mice (*Rorc-cre⁻Fcer1g^{f/f}*, *n* = 7 mice; *Rorc-cre⁺Fcer1g^{f/f}*, *n* = 5 mice) infected with *C. albicans* as in (**A**). **C** Quantification of the fungi load in kidneys of *Rorc-cre⁻Fcer1g^{f/f}* and *Rorc-cre⁺Fcer1g^{f/f}* mice on day 5 post the infection (*n* = 5 mice per group). **D, E** Representative PAS staining images of kidneys of *Rorc-cre⁻Fcer1g^{f/f}* and *Rorc-cre⁺Fcer1g^{f/f}* mice on day 5 post the infection. Arrows: sites of fungal invasion; Boxes: medulla area with mycelia, showing in higher magnification in (**E**). **F** Quantification of the renal pathology scores of images as in (**D, E**) (*n* = 6 mice per group). Data are representative of three (**B**) or two (**C**–**F**) independent experiments shown as the mean ± SEM. Statistical significance was tested by two-tailed *t* test (**C, F**) or log-rank (Mantel-Cox) test (**B**).

that JAK signaling pathway was involved in FcεR1γ-mediated proinflammatory immune response in ILC3s.

## FcεR1γ-mediated antifungal immunity of ILC3s is dependent on IL-17A

To further determine the dependence of IL-22 and IL-17A on FcεR1γ-mediated antifungal immunity in ILC3s, we administrated recombinant IL-17A or IL-22 into CKO mice during the *C. albicans* infection (Fig. 6A). We found that supplementation with IL-17A, but not IL-22, significantly delayed the fatality of CKO mice (Fig. 6B). Additionally, IL-17A treatment significantly reduced fungi burden and pathological damage in the kidneys of CKO mice (Fig. 6C–E). Therefore, the compromised antifungal immunity of FcεR1γ-deficient ILC3s was partially restored by exogenous IL-17A. We also confirmed the systemic reduction of IL-17A in the serum of CKO mice (Fig. 6F). Together, these data indicated that intestinal ILC3s contributed to IL-17A levels in the periphery for host defense against infection.

Next, we aimed to identify the potential co-receptor of FcεR1γ on the membrane of ILC3s for antifungal immunity. Previous studies have suggested that NKp46 is not required for IL-22 production in ILC3 under the steady state condition[39] or during infection[40]. Additionally, we observed that FcεR1γ deficiency impaired cytokine production in CCR6⁺ and CCR6⁻ ILC3s, both of which highly

expressed CD16. Consistently, we found that the absence of CD16, but not NKp46, impaired cytokine production in ILC3s in response to fungi infection (Fig. 7A, B, and Supplementary Fig. S8A, B). Moreover, the protective function of FcεR1γ expression in ILC3s was lost when mice were crossed onto the *Rag1⁻/⁻* genetic background (Fig. 7C), indicating that the function of FcεR1γ in ILC3s relied on components of the adaptive immune system, such as IgG, which is the ligand of CD16. Importantly, the replenishment of mouse IgG significantly delayed the mortality (Fig. 7D) and increased the cytokine production in ILC3s in *Rag1⁻/⁻* control mice compared to CKO mice (Fig. 7E, F, and Supplementary Fig. S8C, D). Collectively, these data demonstrated the crucial role of the IgG-CD16-FcεR1γ axis in ILC3 activation and antifungal immunity. Intriguingly, through an in vitro IgG-binding assay (Supplementary Fig. S9A), we found that while monoclonal IgG antibodies efficiently bound to RAW cells in an FcεR1γ-dependent manner (Supplementary Fig. S9B), these purified antibodies could not bind to ILC3 under these in vitro conditions (Supplementary Fig. S9C). These findings suggested that compared to macrophages, the binding efficiency of IgG on ILC3s was relatively low, and additional components, such as the immune complex composed of antibodies and antigens in vivo, was required for the binding of antibodies on ILC3s.

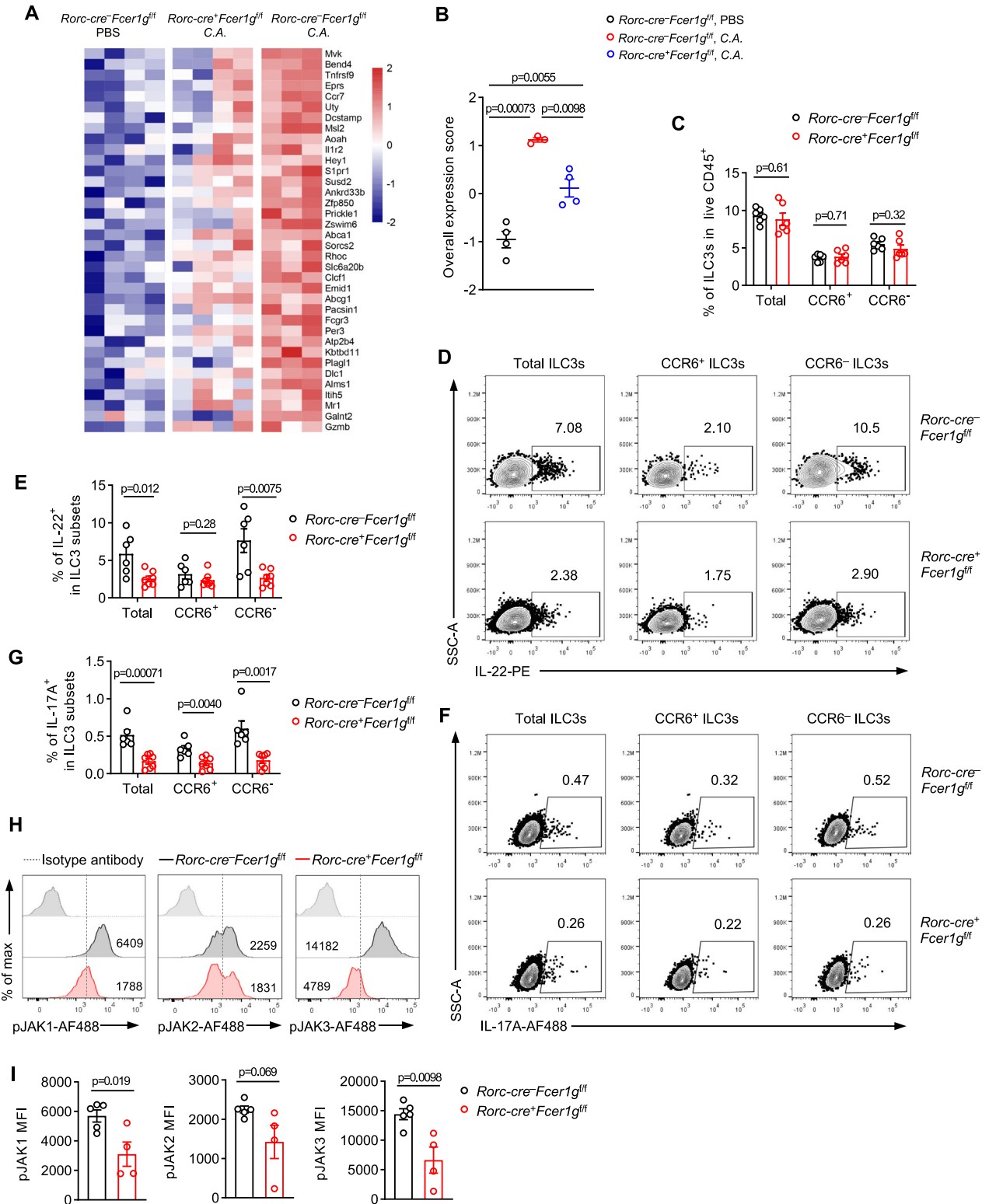

**Fig. 5 | FcεR1γ supports transcriptional states and effector molecules in ILC3s respond to fungal infection. A** Heatmap showing the expression (color bar, Z score) of genes (rows) significantly induced in ILC3s by the fungal infection (Supplementary Fig. S4E) across different genotypes and conditions (columns) (*Rorc-cre⁻Fcer1g^{f/f}* PBS, *n* = 4 mice; *Rorc-cre⁺Fcer1g^{f/f}* *C.A.*, *n* = 4 mice; *Rorc-cre⁻Fcer1g^{f/f}* *C.A.*, *n* = 3 mice). *C.A.* represents *Candida albicans*. **B** Quantification of the overall expression (**Methods**) of the genes in (**A**). **C** Quantification of total, CCR6⁺ and CCR6⁻ ILC3s in the siLPs of *Rorc-cre⁻Fcer1g^{f/f}* and *Rorc-cre⁺Fcer1g^{f/f}* mice on day 5 post the infection (*n* = 6 mice per group). **D-G** Representative flow plots showing the intracellular abundance of IL-22 (**D**) or IL-17A (**F**) in total, CCR6⁺ and CCR6⁻

ILC3s on day 5 post the infection. Quantification of the frequencies of IL-22 (**E,** *Rorc-cre⁻Fcer1g^{f/f}*, *n* = 7 mice; *Rorc-cre⁺Fcer1g^{f/f}*, *n* = 6 mice) or IL-17A (**G,** *Rorc-cre⁻Fcer1g^{f/f}*, *n* = 7 mice; *Rorc-cre⁺Fcer1g^{f/f}*, *n* = 6 mice) -expressing cells in each compartment. **H, I** Representative flow plots showing the phosphorylation of JAK1, JAK2 and JAK3 in total ILC3s from siLPs of indicated mice on day 5 post the *C. albicans* infection (**H**). Quantification of MFI (**I,** *Rorc-cre⁻Fcer1g^{f/f}*, *n* = 5 mice; *Rorc-cre⁺Fcer1g^{f/f}*, *n* = 4 mice). Data are representative of two (**C–G**) or one (**H, I**) or from two (**A, B**) independent experiments shown as the mean ± SEM. Statistical significance was tested by two-tailed *t* test (**B, C, E, G, I**).

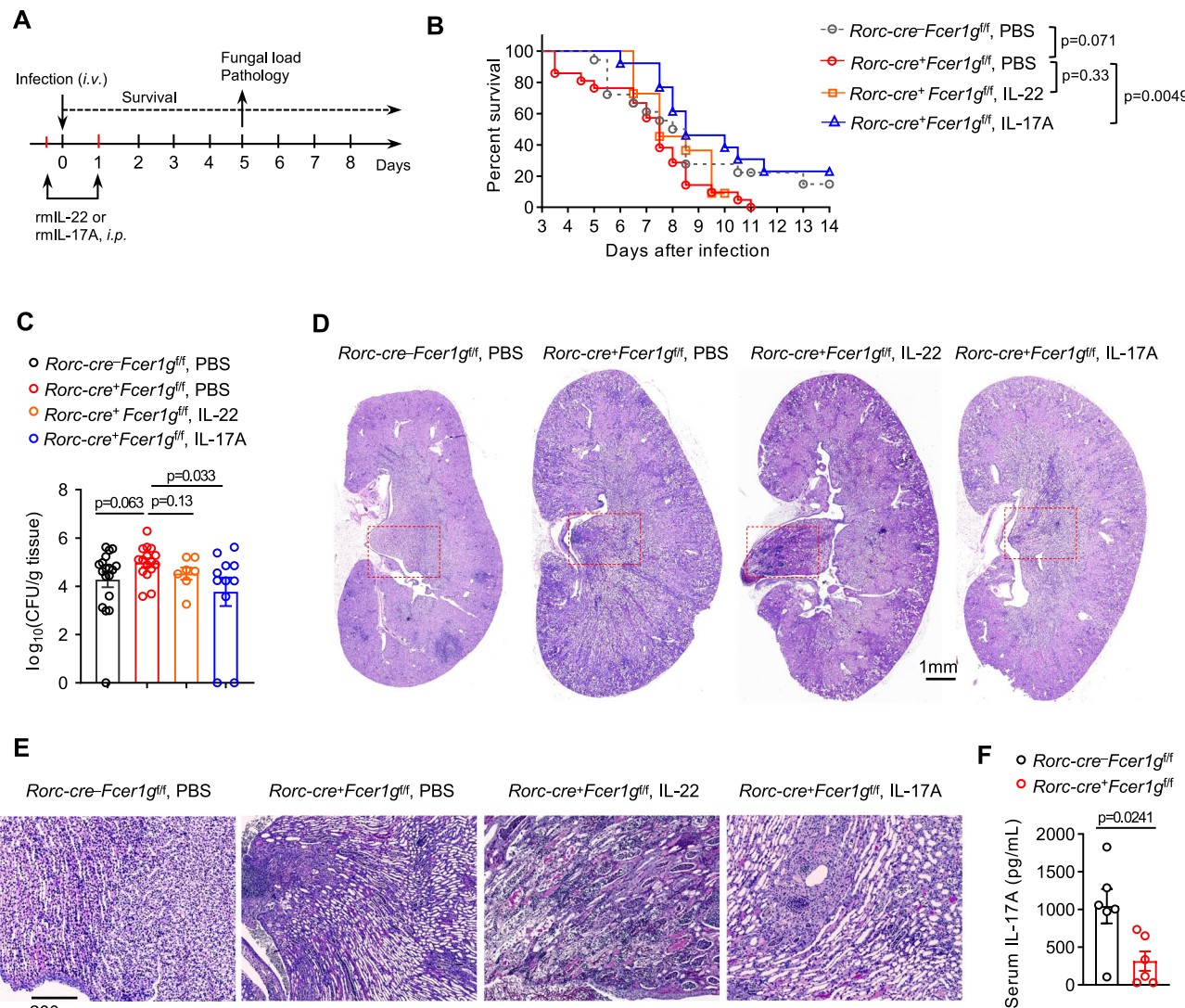

**Fig. 6 | Exogenous IL-17A partially restores compromised antifungal immunity induced by the deficiency of FcεR1γ in ILC3s. A** Experimental scheme. **B** Survival curves of *C. albicans*-infected *Rorc-cre⁺Fcer1g*^f/f^ mice treated with IL-17A, IL-22 or vehicle (PBS) (*Rorc-cre⁻Fcer1g*^f/f^ PBS, *n* = 18 mice; *Rorc-cre⁺Fcer1g*^f/f^ PBS, *n* = 21 mice; *Rorc-cre⁺Fcer1g*^f/f^ IL-22, *n* = 13 mice; *Rorc-cre⁺Fcer1g*^f/f^ IL-17A, *n* = 11 mice). **C** Quantification of the fungi load in kidneys of mice as in (**A**) on day 5 post the infection (*Rorc-cre⁻Fcer1g*^f/f^ PBS, *n* = 18 mice; *Rorc-cre⁺Fcer1g*^f/f^ PBS, *n* = 16 mice; *Rorc-cre⁺Fcer1g*^f/f^ IL-22, *n* = 7 mice; *Rorc-cre⁺Fcer1g*^f/f^ IL-17A, *n* = 11 mice). **D, E**

Representative PAS staining images of kidneys of mice as in (**A**) on day 5 post the infection. Boxes: medulla area with mycelia, showing in higher magnification in (**E**). **F** Quantification of IL-17A in the serum of mice as in Fig. 4A on day 5 post the infection. *n* = 6 mice per group. Data are representative of three (**D, E**) or one (**F**) or are pooled from four (**B**) or three (**C**) independent experiments shown as the mean ± SEM. Statistical significance was tested by two-tailed *t* test (**C, F**) or log-rank (Mantel-Cox) test (**B**).

## Discussion

In this study, we revealed that FcεR1γ played a critical role as an adapter that stabilizes the expression of multiple major phenotypic proteins in ILC3s. More importantly, we demonstrated that FcεR1γ is necessary for the proper function of ILC3s in response to bacterial or fungal infections, and this functionality is dependent on the presence of IgG and CD16.

Our findings highlight the essential role of FcεR1γ in the expression of NKp46 and CD16 in ILC3s. Through an alanine scanning mutagenesis assay using BMDM as a surrogate cellular system, we identified that D29 and L39 in the transmembrane region are essential residues for maintaining the expression of CD64 in BMDMs, NKp46 and CD16 in ILC3s. This observation aligns with the previous report that the association between the transmembrane regions of FcεR1γ and its partners, such as the high-affinity IgE receptor FcεRI and IgA receptor FcαRI, relies on both arginine/aspartic acid charge interaction and leucine zipper-like interaction[41]. Of note, while FcεR1γ is

constitutively associated with common beta-chain of the IL-3 receptor in basophils, its deficiency does not affect the expression of the IL-3 receptor itself[17]. However, FcεR1γ is essential for its transduction of IL-3 signals in basophils, which is dependent on D29 but not L39. The detailed molecular mechanism underlying how D29 and L39 stabilize the binding partners remains unclear. It has been suggested that FcεR1γ may prevent the NKp46 degradation by proteasome in NK cells in vitro[42]. However, further research is needed to investigate whether and how FcεR1γ interacts with proteasome through these two residues in ILC3s.

IL-17−mediated immunity has been shown to be a crucial host defense mechanism against fungal infections[32,43,44]. Intriguingly, recent studies have suggested that IL-17-secreting ILCs, rather than Th17 cells, are primarily responsible for controlling *C. albicans* infection[27]. In our work, we found that the fungal infection potently activates genes involved in the JAK-STAT signaling and cytokine secretion pathways, in ILC3s in a FcεR1γ-dependent manner. It has been reported that during

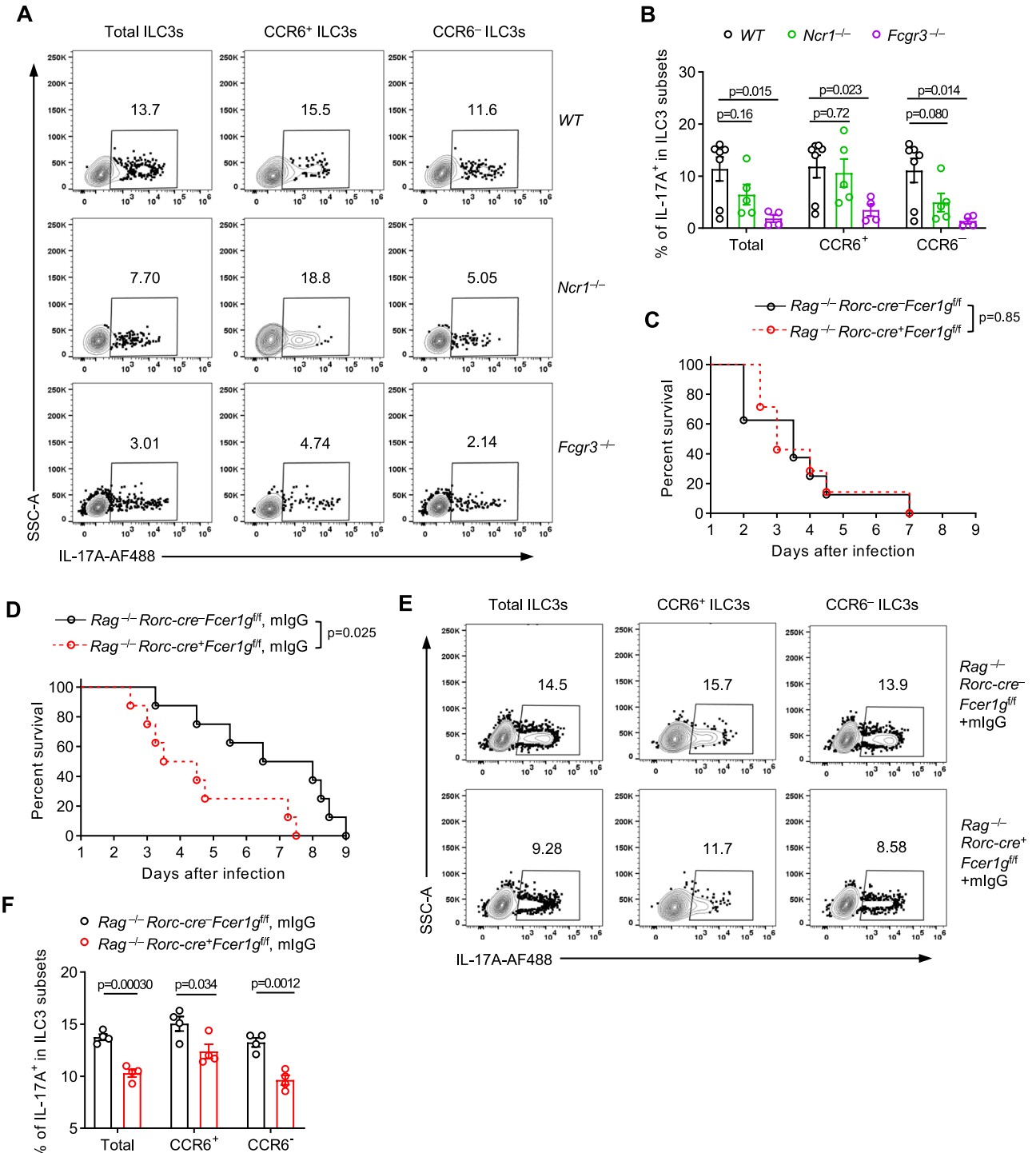

**Fig. 7 | FcɛR1γ-mediated protective immunity in ILC3s depends on the presence of CD16 and IgG antibody. A, B** Representative flow plots showing the intracellular abundance of IL-17A (**A**) in total, CCR6+ and CCR6- ILC3s from siLPs of mice with the indicated genotypes on day 5 post *C. albicans* infection. Quantification of the frequencies of IL-17A-expressing cells in each compartment (**B**, WT, *n* = 7 mice; *Ncr1*-/-, *n* = 5 mice; *Fcgr3*-/-, *n* = 4 mice). **C** Survival curves of *C. albicans*-infected *Rag1*-/-*Rorc-cre*-*Fcer1g*f/f (*n* = 8 mice) and *Rag1*-/-*Rorc-cre*+*Fcer1g*f/f mice (*n* = 7 mice). All mice were intravenously infected with *C. albicans* on day 0 (**Methods**). **D** Survival curves of *C.* *albicans*-infected *Rag1*-/-*Rorc-cre*-*Fcer1g*f/f and *Rag1*-/-*Rorc-cre*+*Fcer1g*f/f mice treated with mIgG (**Methods**) (*n* = 8 mice per group). **E, F** Representative flow plots showing the intracellular abundance of IL-17A (**E**) in total, CCR6+ and CCR6- ILC3s from siLPs of mice in (**D**) on day 5 post the infection. Quantification of the frequencies of IL-17A (**F**)-expressing cells in each compartment (*n* = 4 mice per group). Data are pooled from three (**A, B**) or representative of three (**C**) or two (**D-F**) independent experiments shown as the mean ± SEM. Statistical significance was tested by two-tailed *t* test (**B, F**) or log-rank (Mantel-Cox) test (**C, D**).

*C. albicans* infection, macrophages secrete IL-7, which in turn induces IL-22 production in ILC3s[45]. Therefore, FcɛR1γ might assist ILC3 to in sensing the inflammatory environmental changes induced by infections, thus enabling their participation in the anti-infection immunity.

However, it is worth noting that FcɛR1γ expression in ILC3s does not respond to DSS-induced inflammation, suggesting that environmental changes sensed by FcɛR1γ-mediated mechanism may be specific to the infections.

In our in vivo experiments, we observed that the IgG-CD16 pathway promoted cytokine production and anti-fungal immunity in ILC3s. However, in vitro experiments did not show direct binding of IgG with ILC3s through CD16. This discrepancy may be attributed to the low-to-medium affinity Fc receptors, including CD16, which can only bind IgG when it is present in an immune complex, aggregated, or opsonized by other components[15]. In contrast, high-affinity Fc receptors, such as CD64, can bind free/monomeric IgG. This may explain why CD16 deficiency resulting from FcεR1γ knockout leads to a defect in cytokine production only during infection but not at homeostasis. The specific in vivo mechanism by which CD16 contributes to ILC3 function would be worth of further investigation. Moreover, it's possible that FcεR1γ maintains the expression of unknown receptors that can direct sense and bind pathogens or their derivatives. Further research is needed to elucidate the FcεR1γ-mediated molecular mechanisms at the cell membrane for sensing, as well as the downstream signaling in ILC3s that regulate transcriptional states and cytokine production.

In summary, our study defined a previously unrecognized role of FcεR1γ in maintaining the expression of phenotypic receptors and effector function in ILC3s. This advancement in our understanding on ILC immunobiology and disease pathology may lead to the discovery of new sensing pathways for ILC3 activation and provide new insights for the treatment of infectious diseases.

## Methods

### Mice
C57BL/6 J (Jax 000664), *Rorc-cre* mice (B6.FVB-Tg(Rorc-cre)1Litt/J, Jax 022791) and *Rag1* KO mice (B6.129S7-*Rag1^{tm1Mom}*/J, Jax 002216) were obtained from the Jackson Laboratory. *Fcer1g flox^{tm1c}* mice were produced by crossing *Fcer1g^{tm1a(KOMP)}* allele (MMRRC 047680-UCD) from NIH Knockout Mouse Project (KOMP) with an Flp transgenic mouse strain[46] to remove the FRT-flanked knockout-first cassette. *Rorc-cre Fcer1g* flox mice were generated by mating *Rorc-cre* mice with *Fcer1g flox^{tm1c}* mice. *Rag1^{−/−} Rorc-cre Fcer1g* flox mice were generated by mating *Rag1* KO mice with *Rorc-cre Fcer1g* flox mice. *Fcer1g* KO (*Fcer1g* Germline KO, NM-KO-190187) mice were purchased from Shanghai Model Organisms Center, Inc.. *Ncr1* KO mice and *Fcgr3* KO mice were generated as descried[47] at the Center for Excellence in Molecular Cell Science, Chinese Academy of Sciences. Briefly, the mRNA of hA3A-eBE3-Y130F, *Ncr1*-sgRNA (5′-GAGTATCAACTGTACTTTGA-3′) and *Fcgr3*-sgRNA (5′- ATGTTTCAGAATGCACA CTC-3′) were transcribed and purified with the MEGAclear Transcription Clean-Up Kit (Invitrogen). The mixture of hA3A-eBE3-Y130F mRNA (100 ng/μL) and sgRNA (100 ng/μL) was injected into the cytoplasm of zygotes (C57BL/6 J) using a micromanipulator (Olympus) and a FemtoJet microinjector (Eppendorf). The obtained embryos were cultured in AA-KSOM (Merk) medium at 37 °C, 5% $CO_2$ for 18–24 h. Then, 2-cell embryos were transferred into the oviducts of pseudopregnant ICR females. F0 mice carrying the expected premature stop codon of targeted genes were selected by Sanger sequencing and then the deletions of the protein were confirmed by FACS analysis. All mice were housed in specific-pathogen-free conditions at 22–24 °C with a humidity of 40%–70% and a light/dark cycle of 12 h. Euthanasia was performed using cervical dislocation or $CO_2$ asphyxiation. All animal experiments were approved by the Institutional Animal Care and Use Committee (IACUC) of Westlake University under protocol # 18-002-XHP.

To induce colitis, 8 to 12 weeks old male mice were treated with 3.0% (w/v) of dextran sulfate sodium (DSS) in drinking water for 7 days and then were switched to sterile distilled water.

To establish invasive fungal infection, *C. albicans* (strain SC5314) were inoculated in YPD medium and cultured at 37 °C, 200 rpm for 12–16 h. Then the fungi were washed, diluted and coated in YPD-Agar plates for CFU counting. A total of 1.8 or $1.5 \times 10^5$ CFU or $1.5 \times 10^6$ CFU *C. albicans* when indicated, were *i.v.* injected to age and gender matched mice.

To establish *C. rodentium* infection, 5 to 6 weeks old mice were fasted for 8–12 h and then infected with $3–5 \times 10^9$ CFU (females) or $1.2–1.4 \times 10^{10}$ (males) *C. rodentium* (strain ICC180, kanamycin resistant) by oral gavage. The stools were collected, resuspended, and coated in LB-Agar plates with kanamycin (50 μg/mL) for CFU counting.

### Isolation of immune cells from Peyer's patches (PP) and lamina propria (LP)
PPs were dissected from the small intestine, pierced with a fine forcep. Tissues were digested in fresh RPMI-1640 medium containing 100 μg/mL Liberase TM and 50 μg/mL DNase I at 37 °C on a roto-mixer for 15 min. The medium was collected and added to ice-cold MACS buffer (pH 7.4; PBS with 2% FBS and 2 mM EDTA). The remaining tissues were digested for another 15 min. Supernatants from the two steps were combined and passed through 70 μm strainer for further use.

To isolate cells from LP, the intestines were opened longitudinally and washed in ice-cold PBS. Then cut the intestines into 0.5–1.0 cm fragments after remove fat and PPs. Epithelial cells and intraepithelial lymphocytes (IEL) were dissociated by rotation (350 rpm) in pre-warmed (37 °C) PBS containing 5% FBS, 10 mM EDTA and 1 mM DTT at 37 °C for 30 min. After washing with PBS containing 5% FBS, tissues were digested in pre-warmed (37 °C) RPMI-1640 medium with 10%FBS, 1 mg/mL Collagenase IV (1.5 mg/mL for colon), 20 μg/mL DNase I, 10 mM HEPES, 1 mM $Ca^{2+}$ and 1 mM $Mg^{2+}$ with 550 rpm rotation for 20 min at 37 °C. The supernatants were then passed through 70 μm strainer after added to 2 mM EDTA. Centrifuge at 4 °C, $500 \times g$ for 5 min. Then the pellets were resuspended in 3 mL 40% Percoll, loaded on 1 mL 80% Percoll, and centrifuged at 25 °C, $600 \times g$ for 20 min. Cells at the interphase of the two layers were collected and washed in MACS for counting, staining, culture and/or sorting.

### Flow cytometry and cell sorting
After blocking nonspecific binding with Fc-blocking antibody 2.4G2 by incubation for 10 min on ice, single-cell suspensions were stained with surface-labeled antibodies (see Supplementary Table 1) for 20–30 min at 4 °C and then were washed with MACS buffer. In the experiment of detecting CD16 (*Fcgr3*), we used FBS instead of 2.4G2 for blocking.

For intracellular staining of transcription factor, the LPLs were fixed and permeabilized with Fixation/Permeabilization buffer (Foxp3 Transcription Factor Staining Buffer Kit, eBioscience) after surface markers staining. For intracellular staining of FcεR1γ, BMDMs or LPLs were fixed and permeabilized with Fixation/Permeabilization Solution (BD) after surface markers staining, subsequently stained with anti-FcεR1γ antibody for 1 h at 4 °C.

For intracellular staining of IL-22 and IL-17A, sorted CD45+live SSC^{low} LPLs (Purity > 92%) were stimulated with Cell Stimulation Cocktail, Protein Transport Inhibitor Cocktail (eBioscience) and 40 ng/mL rmIL-23(R&D Systems) in complete RPMI medium (10% FBS, 10 mM HEPES, 1 mM sodium pyruvate, 80μM 2-mercaptoethanol, 2 mM glutamine, 100 U/mL penicillin, 100 mg/mL streptomycin) for 2.5 or 4 h when indicated at 37 °C, 5% $CO_2$. After stimulation, cells were stained with the surface-stained markers, subsequently fixed and permeabilized with Fixation/Permeabilization Solution. After permeabilization, the cells were incubated for overnight at 4 °C with the indicated anti-cytokine antibodies.

For analysis of phosphorylation of JAKs, the LPLs were stimulated with 100 ng/mL rmIL-23 (R&D Systems) and rmIL-7 (BioLegend) in complete RPMI medium for 40 min at 37 °C, 5% $CO_2$. After stimulation, cells were stained with the surface-stained markers, subsequently fixed and permeabilized with pre-cooled methanol, then the cells were incubated for overnight at 4 °C with the indicated antibodies.

Flow cytometry was performed with CytoFLEX (Beckman Colter) or LSR Fortessa (BD) and analyzed with FlowJo software v10.0.7 (TreeStar).

For sorting, single-cell suspensions were sorted to >97% purity after surface markers staining using a FACSAria Fusion (BD) or FACSAira III (BD) sorter with a 70μm Nozzle at 4 °C.

## Quantitative real-time PCR

500–2000 freshly sorted cells (Purity > 98%) were lysed in 10 μL of TCL buffer (Qiagen) containing 1% 2-Mercaptoethanol (Sigma). Full-length cDNA was generated and amplified using SMART-Seq2 protocol as described previously[48]. Quantitative PCR was performed on a qTOWER384G System (Jena) with iTaq Universal SYBR Green Supermix (Bio-Rad). Each reaction was performed with three replicates. Primers used for quantitative PCR are in Supplementary Table 2.

## Lentivirus production and infection with BMDMs, RAW cells and ILC3s

To overexpress WT or mutated *Fcer1g* in BMDMs, the coding sequence of mouse *Fcer1g* (NM_010185.4) or mutant sequences (Primers used for mutated *Fcer1g* are in Supplementary Table 2) were inserted into pLenti-CMV-RFP-BSD vector (Public Protein/Plasmid Library) and co-transferred into 293 T cells with psPAX2 (Addgene, no. 12260) and pMD2.G (Addgene, no. 12259) using PolyJet (Signagen Laboratories). Supernatant was harvested at 48 h and 72 h and concentrated by ultracentrifuge. Bone marrow cells from *Fcer1g* KO mice were cultured with 20 ng/mL mouse M-CSF (PeproTech) for 7 days to induce BMDMs. At day3.5, BMDMs were infected by lentivirus with target gene and the infection efficiency and protein expression were detected by FACS 3 days after infection.

To overexpress WT or mutated *Fcer1g* in ILC3s, the coding sequence of mouse *Fcer1g* (NM_010185.4) or mutant sequences were inserted into pHAGE-EF1a-IRES-ZsGreen (Addgene, no. 114008), and co-transferred into 293 T cells with psPAX2 and pMD2.G. Supernatant was harvested and concentrated. LPLs from *Rag1*$^{-/-}$ *Rorc-cre*$^+$ *Fcer1g*$^{f/f}$ were isolated and total ILC3s were sorted as CD45$^+$live Lin$^-$ CD90.2$^{high}$ CD45$^{low}$ KLRG1$^-$ cells. Then ILC3s were cultured with rmIL-2, rmIL-7 (all 20 ng/mL, all from BioLegend) in complete RPMI medium for 3–5 days at 37 °C, 5% CO$_2$. Then ILC3s were infected by lentivirus with target gene and the infection efficiency and protein expression were detected by FACS 2 days after infection.

To knockout the *Fcer1g* in RAW cells, sgRNA (Oligo sequences are in Supplementary Table 2) were inserted into pLenti-CRISPR-v2-GFP vector (Addgene, no. 82416) and lentivirus were harvested as described above. Single-cell clones of infected RAW cells were sorted (GFP$^+$) at day 5 post lentivirus infection and the knockout of *Fcer1g* were confirmed by FACS analysis for FcεR1γ and CD64 expression at day 12 post infection.

## IgG binding assay in vitro

WT, *Fcer1g*$^{-/-}$ RAW cells or fresh lymphocytes of LPs from WT mice were incubated with or without Fc-blocking antibody 2.4G2 for 20 min at 4 °C, followed by incubated with mouse anti-OVA IgG isotypes (1 μg/mL) for 60 min at 4 °C. Then the cells were washed 2 times and stained with surface-labeled antibodies and fluorescent secondary antibodies (donkey anti-mouse IgG, see Supplementary Table 1) for 40 min at 4 °C. After wash, the cells were analyzed by FACS for IgG binding.

## Bioluminescence imaging (BLI)

For BLI, mice were depilated using a shaver and anaesthetized with isoflurane inhalation. All animals were imaged using PhotonIMAGER OPTIMA systems (Biospace Lab) under gaseous anesthesia with isoflurane. The signals were acquired by PhotoAcquisition (Biospace Lab) software and the data were analyzed by M3Vision (Biospace Lab) software.

## Recombinant cytokine or antibody treatment

Mice were injected intraperitoneally with a dose of 1 μg rIL-17A protein (R&D Systems) or 0.8 μg rIL-22 protein (R&D Systems) 6 h before *C.*

*albicans* infection, followed by a booster dose of the same cytokine at 24 h after infection. In parallel, control mice were injected solely with equivalent vehicle.

For antibody treatment, mice were injected intraperitoneally with 200 μg mouse non-specific immunoglobin G (YEASEN) from WT mice 1 day before *Candida albicans* infection, followed by a booster dose 3 days after infection.

## Enzyme-linked immunosorbent assay (ELISA)

To determine the levels of IL-17A in the serum, mouse blood was collected, centrifuged, and the supernatants were collected. Levels of IL-17A in the supernatants were measured by using ELISA kit (BioLegend) according to the manufacturer's instructions.

## Immunoprecipitation-Mass Spectrometry (IP-MS)

The coding sequence of mouse *Fcer1g* with *Flag* sequence at 3' end was cloned into pHAGE-EF1a-IRES-ZsGreen, and co-transferred into 293 T cells with psPAX2 and pMD2.G. Viral supernatant was harvested, concentrated and added to the RAW 264.7 cells for infection (Empty virus was as control). GFP$^+$ cells were sorted 2.5 days after infection. IP-MS assays were performed with standard protocols. In brief, $1 \times 10^7$ GFP$^+$ RAW cells were washed and homogenized in lysis buffer (50 mM Tris-HCl (pH 7.4), 150 mM NaCl, 1% Triton X-100, 5 mM EDTA, and proteinase inhibitor cocktail (Roche)). Anti-FLAG antibodies (Sigma-Aldrich) were added to the cell lysates and incubated for 4 h at 4 °C, followed by incubation with Protein A/G Magnetic Beads (Thermo) for 2 h at 4 °C. After washed with washing buffer (50 mM Tris-HCl (pH 7.4), 500 mM NaCl, 0.1% Triton X-100, 5 mM EDTA, and proteinase inhibitor cocktail), IP pellets were diluted in SDS-loading buffer and then analyzed with SDS-PAGE. After pre-treatment, IP products were analyzed by Orbitrap Exploris™ 480 (Thermo). The raw MS data were processed by MaxQuant for further quantitative analysis.

## Histological analysis

Kidneys from *C. albicans* infected mice and colons from *C. rodentium* infected mice or DSS treated mice were fixed for at least 24 h with 4% paraformaldehyde and were embedded in paraffin. Cross-sectional tissues were stained with Periodic acid–Schiff (PAS) or hematoxylin and eosin. Pathological score was evaluated by an observer masked to treatment group. Pathological scores of colon tissue were assessed with a previously described scoring system: 0=no evidence of inflammation,1=low level of inflammation with scattered infiltrating mononuclear cells (1–2 foci), 2=moderate inflammation with multiple foci, 3=high level of inflammation with increased vascular density and marked wall thickening, 4=maximal severity of inflammation with transmural leukocyte infiltration and loss of goblet cells. Intestinal crypt elongation was also recorded in *C. rodentium* infected mice. The extent of kidney histological damage was graded on a scale of 0 to 10, where 0–1 indicates no visible renal parenchymal damage, 2–4 indicates focal mild inflammation, 5–6 indicates patchy moderate inflammation with focal renal parenchymal damage, and 7–8 indicates extensive inflammation with severe renal parenchymal tissue destruction, 9–10 indicates extensive inflammation with large number of fungal hyphae. In addition, each kidney was evaluated for the extent of renal cortical and medullary involvement by using the same scoring criteria.

## Bulk RNA-seq

1000–2000 freshly sorted cells (Purity > 98%) from each group with at least three replicates were lysed in 10 μL TCL buffer with 1% 2-Mercaptoethanol. Libraries were prepared with SMART-Seq2 protocol. Briefly, total RNAs were extracted by $2.2 \times$ RNA-SPRI beads (Beckman Colter) and reverse transcribed by Maxima H Minus Reverse Transcriptase (Thermo). The cDNA were amplified with 13 cycles by KAPA HiFi HotStart ReadyMix(KAPA Biosystems) and the whole

transcriptome amplified product were purified by 0.7 × DNA-SPRI beads (Beckman Colter). Libraries were processed with TruePrep DNA Library Prep Kit V2 (Vazyme) and paired-end sequenced (150 bp×2) with HiSeq X (Illumina).

Raw fastq files were trimmed using cutadapt (v2.10) with '-q 30' to filter for Q30 reads and cut possible adapter sequence or bad-quality bases on the 3' primer of each reads. Trimmed reads were aligned to the mouse genome/transcriptome (GENCODE GRCm38 vM25, mm10) using STAR (v2.7.5c) with 'twopassModeBasic', expression abundances were estimated (expected-counts and TPM) using RSEM (v1.3.3). The counts matrix output from gene-results were processed with the edgeR package (v3.32.0) in R to analyze differential gene expression with default parameters. Only protein-coding genes that had CPM > 2 in at least one tissue/condition were calculated. GSEA(v4.2.1) were performed using the official tool with filtered edgeR (v3.32.0) normalized (TMM) expression matrix, and results were replotted in R.

### Droplet-based scRNA-seq library construction

Single cells were captured via the Chromium Next GEM Single Cell 3′ Reagent Kits v3.1 (10 × Genomics) according to the manufacturer's protocol. Briefly, freshly sorted murine intestinal CD45⁺live Lin⁻ cells (Lin⁻ refers to CD3e⁻CD19⁻CD5⁻Gr-1⁻CD11b⁻CD11c⁻TCRβ⁻TCRγ/δ⁻ FcεRIα⁻), which were prelabeled with hashtag antibodies (BioLegend), were pooled and loaded at 36,000 or 44,000 cells (about 5500 to 6000 cells per sample) per channel. The cells were then partitioned into the GemCode instrument, where individual cells were lysed and mixed with beads carrying unique barcodes in individual oil droplets. The products were subjected to reverse transcription, emulsion breaking, cDNA amplification, and sample index attachment. Libraries were pair-end (150 bp×2) sequenced with NovaSeq (Illumina).

### scRNA-seq data processing

Reads demultiplexing, alignment to the mm10 transcriptome and unique molecular identifiers (UMIs) counting/collapsing were using the CellRanger toolkit (v6.1.1, 10x Genomics). Main downstream analysis like 'Demultiplexing with hashtag oligos (HTOs)', clustering and filtering were performed in R (v4.0.3)/Rstudio with Seurat (v4.1.0). We applied two runs to get final clusters of pure ILCs. Specifically, in first run, HTO singlets were extracted from FB (Feature Barcode) assay, initial filtering for GEX (Gene Expression) assay was using 'percent.mt <10' and 'nFeature_RNA: 200-4,000'. UMI counts in dataset were then normalized and scaled. We did variable feature selection, dimension reduction, and pre-clustering with ranged parameters, and found markers for each cluster with 'MAST' model. After manually checking the clustering results, more appropriate cutoffs were determined for downstream: PCs 1-24, k.param 20, method 'igraph', resolution 1. We first chose top 2000 highly variable, but then mt.genes, cc.genes and several putative contamination genes (contam.genes) were excluded in dimension reduction step. Those contam.genes were most TCR/BCR genes or stress-related which would often mislead the clustering, and fitered by matching:

"^Tra|^Trb|^Trg|^Trd|^Tcr|^Igm|^Igh|^Igk|^Igl|Jchain|Mzb1| ^Hsp|^Rps|^Rpl|Hbb-|Hba-|^Dnaj|^Fos|^Jun|^AY|^Gm|^Hist|Rik$".

Pre-Annotation were done by checking these markers: Immune (*Ptprc*), T cell (*Cd3d, Cd3e*), NK (*Klrb1c, Tbx21, Ncr1, Eomes*), ILC1 (*Klrb1c, Tbx21, Ncr1, Ifng*), Ccr6⁻ILC3 (*Rorc, Cd93, Il22, Ncr1*), Ccr6⁺ ILC3(*Rorc, Cd93, Il22, Ccr6*), ILC2 (*Gata3, Klrg1, Calca, Il13, Il4, Il5*), B cell (*Cd19, Cd79a, Ms4a1, H2-Aa, H2-Eb1*), plasma cell (*Mzb1, Sdc1, Xbp1*), basophil (*Gata2, Il6, Cpa3, Ms4a2*), stromal cell (*Pdgfra, Dcn, Col1a1, Col1a2, Acta2*).

Possible doublets were tested by DoubletFinder (v2.0.3) with specified ratio 0.05/0.10. In second run, only certain ILCs (NK, ILC1/2/3) and 'DoubletFinder0.10 Singlets' were kept. Advanced filtering was performed using more stringent cutoffs depending on the distribution:

percent.mt < 7.5; nFeature_RNA: 800-3,200; nCount_RNA: 1,600–10,000. Next we did re-clustering with parameters: variable 1500 (also filtered), PCs 1-28, k.param 50, method 'igraph', resolution 1.2. Then we got 12 new clusters, in which one was NK, two were ILC1 and two/two were Ccr6⁻/⁺ ILC3, the rest five were ILC2. In general, colon (LI) and small intestine (SI) sourced ILCs were quite distinct from each other while the main cell types of ILCs were clearly grouped. Cell composition for cell types/tissues/conditions was counted. For the same tissue, DEGs between control and CKO in each main cell type were calculated using findmarkers with 'MAST' model.

### MS data processing

The raw mass spectrometry data were first analyzed using MaxQuant[49] and the "LFQ.intensity" columns (label-free quantification) in the resulting "proteinGroups.txt" file were used as input for the downstream analysis. The quantification matrix output was processed with the DEP package (v1.25.0) in R for data preparation, filtering, variance normalization and imputation of missing values, as well as statistical testing of differentially enriched proteins with default parameters. Only proteins identified in 3 out of 4 replicates of at least one condition were calculated. Enrichment analysis for gene ontology and visualization of functional enrichment result were performed using topGO (v2.54.0) and enrichplot (v1.22.0) respectively, and results were replotted in R.

### Statistics and reproducibility

GraphPad Prism 8.02 software was used for all statistical analysis (except for RNA sequencing data or MS data) and data are presented as mean ± SEM, as indicated in the legend of each figure. Survival curves were analyzed according to the Kaplan-Meier estimator, and the difference between two groups was determined by the log-rank (Mantel-Cox) test. Statistical differences for other experiments were determined by Mann-Whitney U-test or unpaired Student's *t* test as indicated. *p* values of ≤0.05 were considered to represent means with a statistically significant difference. No statistical method was used to predetermine sample size. No data were excluded from the analysis. The investigators were not blinded to allocation during experiments and outcome assessment except for pathological score.

### Reporting summary

Further information on research design is available in the Nature Portfolio Reporting Summary linked to this article.

## Data availability

The transcriptomic sequencing data are available at Gene Expression Omnibus (GEO) under accession number GSE256409. The mass spectrometry proteomics data have been deposited to the ProteomeXchange Consortium via the iProX partner repository under the dataset identifier PXD049222. All other data are available in the manuscript or the supplementary materials. Source data are provided in this paper.

## Code availability

The code used in this manuscript is available from the corresponding author by request.

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

## Acknowledgements

We thank Guanghua Huang (Fudan University) for providing *Candida albicans*; Xu Li (Westlake University) for technical help; all the members of the Xu and He laboratories for discussion and suggestions; the Laboratory Animal Resources Center, High-Performance Computing Center, Flow Cytometry Core and Genomic Core in Westlake University for their support. The schematic diagrams in Figs. 2, 3, and 4A, and Supplementary Figs. S3A, S7A, and S9A were created with BioRender.com and released under a CC-BY-NC-ND license. This work was supported by the National Key R&D Program of China (grants 2019YFA0802900 and 2020YFA0804200 to H.X.), National Natural Science Foundation of China (grants 31900668 to C.H.; 82325023 and U20A20346 to H.X.), the Research Funds of Hangzhou Institute for Advanced Study, and UCAS (grants B04006C019019 and B04006C01600508 to C.H.; B04006C010003 to J.L.), "Pioneer" and "Leading Goose" R&D Program of Zhejiang (2024SDXHDX0001 and 2024SSYS0031 to H.X.) and the Education Foundation of Westlake University.

## Author contributions

C.H. and H.X. conceived and supervised this study. C.H. and W.Z. performed most of the experiments and analyzed the data. Q.L. and Y.L. constructed transgenic mice and performed animal experiments. Y.L., X.C., and L.Z. performed in vitro overexpression experiments and analyzed the data with guidance from H.X. and C.H. S.L., D.C., and C.H. performed computational analysis. F.G. and S.F. conducted some of the animal experiments. D.H. and J.L. provided scientific advice. H.X. and C.H. wrote the manuscript with input from all authors.

## Competing interests

The authors declare no competing interests.
