## [Peer Review File · Nature Communications]

Antibody Fc-Receptor FcεR1γ stabilizes cell surface receptors in group 3 innate lymphoid cells and promotes anti-infection immunityREVIEWER COMMENTS

Reviewer #1 (Remarks to the Author):

The manuscript by Huang et al. entitled "Antibody Fc-receptor common gamma chain FcεRIγ stabilizes cell surface receptors in group 3 innate lymphoid cells and promotes anti-infection immunity" seeks to investigate the role of the FcεRIγ chain in ILC3 functions. ILC3s appear to be able to sense the environment through the integrative reception of different environmental signals. In this context, the analysis of new potential receptors is an important point to evaluate ILC functions. Here, the authors provide interesting observations on the potential role of FcεRIγ+ ILC3s under steady state and inflammatory conditions. However, while the mouse model used seems relevant and well-constructed, as reflected in the beginning of the paper, I am not convinced by the rest of the paper. The key limitations of the study in my opinion are:

- The lack of clearly identified co-receptor(s) in NKp46+ and CCR6+ ILC3s that can explain the observed phenotypes.
- The lack of a functional mechanism associated with the FcεRIγ in ILC3s.
- The relevance of in vitro models used to justify the role of FcεRIγ in NKp46/CD16 expression in ILC3s
- The lack of a complete analysis of the other RORγt+ cells in steady state and inflammatory conditions.
- The relevance of the *C. Albicans* model used to analyze the function of intestinal ILC3s.
- The lack of a convincing discussion.

For these reasons, the paper appears too preliminary to be published in 'Nature Communications'. However, I encourage the authors to explore this further. Details are provided below.

Figure1

It is unclear whether CD16 is expressed in ILC3 subsets. CCR6+ and CCR6- ILC3s do not appear to express CD16 (Fig.1G), in contrast to the appearance of CD16 expression (Fig.1A/B/C). In addition, CCR6+ do not appear to express Fcgr3 transcripts (Fig.1J). What is the expression of CD16 on CCR6+ ILC3s? The authors need to clarify this point.

If all ILC3s express CD16, what is the role of this receptor? To my knowledge, this role has not been described in the literature, and the authors intend that CD16 contributes to ILC3 function, but how?

Furthermore, if CD16, like NKp46, is not expressed on the surface of CCR6+ ILC3s, what is/are the co-receptor(s) for FcεRIγ on CCR6+ ILC3s? and what is/are their role(s)?

CD16 is the type III Fcγ receptor that binds the Fc portion of IgG antibodies. Did the authors test this feature? Did the authors test the functions of FcεRIγ+ ILC3s in the absence of adaptive cells? Knowing that ILC3s have a more important functionality in Rag1/2-/- mice, and thus in the absence of B cells and IgG, what would be the role of CD16 in this case?

Minor comment: The figure 1B needs to be revised. The ILC3 subsets are not clearly visible (CCR6+/-) and the control isotype is missing (bottom panel).

Figure2

Figure 2A is misleading. It may give the impression that ILC3s were used in this experiment, but a HEK model was used. This model appears to be useful for studying the importance of certain residues, but not for verifying the maintenance of receptors in ILC3s. To fully understand their role in NKp46 and CD16 expression, it would be more appropriate and comprehensive to study ILC3s or a similar model.

Figure3

The authors observed a very low percentage of IL-22 producing cells after stimulation. (Figure 3F-I). From different publications (and different labs), it appears that the percentage of IL-22 in ILC3s is around 25-40% (Zhou et al. Nat. Immunol. 2022; Giacomini et al. JEM 2019 ; Talbot et al. Nature 2020). Therefore, stimulation does not seem to be appropriate or effective in these experiments. The low percentage of IL-17+ ILC3s and the lack of differences between CCR6+ and

CCR6- ILC3s are also 'suspicious' for the optimal cytokine detection. Further experiments are necessary as low cytokine detection may interfere with the interpretation of the results. Indeed, I note that with optimal cytokine detection, the authors did not observe any differences between WT and KO mice (Figure S3E-G).

How do the authors explain the phenotype of *RorcCre+Fcer1f/f* mice with *C. Rodentium*? How can the FcεRIγ receptor cause this defect in cytokine production?

The authors do not describe an associated receptor in CCR6+ ILC3s. So how can FcεRIγ block the production or secretion of IL-22 and IL-17? Similarly, what is the mechanism in NKp46+ ILC3s? The authors can argue a role for NKp46. However, Wang et al. (Plos Biol 2018) have already investigated the potential role of NKp46. Using an NKp46 deficiency model (*Ncr1gfp/gfp* mice), they demonstrated that NKp46 is required for ILC1 development, but does not affect the ILC3 development and their IL-22 production. Satoh-Takayama et al. showed also that NKp46 is not required for IL-22-mediated ILC3 in the gut to defend against *C. Rodentium* (J. Immunol 2009), so how can the authors explain the observed phenotype in the absence of a clearly identified co-receptor?

Can the authors explain the difference in phenotype of ILC3s with CR and DSS? Is the FcεRIγ not 'engaged' in the DSS model?

This study shows that T cells are not affected in *RorcCre+Fcer1f/f* mice under steady-state conditions. However, the effect of the *Fcer1g* deletion on inflammatory T cells in the DSS, *C. Rodentium* and *C. Albicans* models remains unknown. The authors cannot exclude an expression of FcεRIγ in inflammatory RORγt+ T cells or IELs, which may influence the inflammatory response and the development of infection. In a recent study, Chou and colleagues defined a distinct population of innate like T cells with cytotoxic potential, called αβILCTCKs, that exhibit reactivity to tumor antigens (Chou et al. Nature 2022). They demonstrated that this population is replenished by thymic progenitors and infiltrates the tumor such as pancreatic cancer or colorectal cancer tissue. We already know that the T cell population is composed of a large diversity of resident, activated and memory cells that are sensitive to the environmental cues and are plastic (Ribot et al Nat. Rev. Immunol. 2021; McDonald et al. Nat. Rev. Immunol. 2018; Caruso et al. Nat. Rev. Immunol. 2020). As observed in the tumor model, FcεRIγ+ RORγt+ T cells could appear during *C. Rodentium* (or *C. Albicans*) infection and affect the intestinal barrier. Due to the crucial role of the T cell response during the *C. rodentium* infection (Stockinger Curr. Opin. Microbiol. 2021), the role of RORγt+ T cells in *RorcCre+Fcer1f/f* mice should be addressed.

Minor comment: In Fig.3 F-I, the authors should indicate the organ used for these experiments (SI or Colon?)

Figure 4/5

The authors chose to administer *C. Albicans* by intravenous injection (Fig4.A). This choice seems somewhat surprising given that the analysis focuses on intestinal ILC3s and not peripheral ILC3s. Did the authors verify the presence of *C. Albicans* in the gut? Furthermore, the authors focused on pathological signs in the kidney (Fig.4 C-F), but what is the phenotype and activation of ILC3s in this organ? As mentioned by the authors, ILC3 may be a source of TH17 cytokines during the early stages of infection (Gladiator et al. J Immunol 2013). However, another group, using *Il17a* reporter mice, reported that TH17 and γδ T cells are the main source of IL-17 (Ramani et al. JCI 2018). While ILC3s (with redundant function with T cells) have been described in the infected tongue (by oral gavage; Break et al. Science 2021), *Il17YFP+* ILC3 were not detected in either sham or *C. Albicans*-infected kidneys (Ramani et al. JCI 2018).

Based on this route of transmission, it can be assumed that *C. Albicans* is able to infect all organs independently of the intestine. Therefore, intestinal ILC3s alone would be able to interfere with the infection in the periphery? How could this happen?

The authors studied the transcriptional state of ILC3s in the gut after infection, but the lack of analysis of *RorcCre-Fcer1f/f* and *RorcCre+Fcer1f/f* ILC3s in steady-state conditions is detrimental

to interpretation (Fig5A, S4E, S5C, S5E).

As mentioned above, TH17 cells have an important role in the host protection, what is the phenotype and function of intestinal and renal T cells in *C. Albicans* infected *RorcCre-Fcer1f/f* and *RorcCre+Fcer1f/f* mice?

As noted above, the production of IL-22 and IL-17 does not seem to be in agreement with observations from other labs (Fig5D-F). It seems necessary to re-evaluate this part of the experiment in order to best interpret the results.

Reviewer #2 (Remarks to the Author):

The manuscript "Antibody Fc-receptor common gamma chain 1 $Fc\epsilon R1\gamma$ stabilizes cell surface receptors in group 3 innate lymphoid cells and promotes anti-infection immunity" by Huang et al described the role of the common Fc epsilon receptor gamma chain in a select mucosal immune cell group. In general, the manuscript is well written and the data generally compelling. It is not surprising that removing this protein from a cell limits surface expression of Fc gamma receptor 1 and 3a (in humans) and other receptors; this is well known. Thus, the novelty lies not in the reduced surface expression resulting from the knockout, but rather in the impacts associated with these knockout. The authors demonstrate that a common colon damage model using DSS is not affected by the knockout, but responses to bacterial and fungal pathogens are. A major objection to the conclusions drawn results from the use of the word "regulates." $Fc\epsilon R1\gamma$ cannot be seen as a regulator unless its expression changes naturally, and thus imparts a change on other interacting proteins. The authors present no evidence supporting this assertion. Rather, $Fc\epsilon R1\gamma$ is necessary for the surface expression of select receptors that mediate select responses. All occurrences of interpretations relating to regulation should therefore be revised.

A few other comments:

Do ILC3s express CD3zeta, which can also substitute for $Fc\epsilon R1\gamma$ to promote CD16a surface localization? Low expression may explain the residual protein levels in some of the knockouts (Figure 1G, for example).

Figure 4D. it is unclear how the sites of fungal invasion are chosen, they appear randomly chosen to this naïve reviewer.

Line 98, it would be helpful to describe the justification for choosing the *Rorc* promoter to drive cre expression, and introduce the caveats described later.

Line 109. "Completely diminished" is an odd phrase. Eliminated or significantly reduced would be more precise.

The alanine scanning experiments using BMDM are only tangentially related to the core of the paper, that $Fc\epsilon R1\gamma$ knockout affects multiple functions. The identification of D29 doesn't impact the more impactful experiments, including the bacterial and fungal infection models. This section could be removed to improve conciseness.

Line 155. "In consistent," is unclear

Line 173. "The distinct requirement of $Fc\epsilon R1\gamma$ in ILC3s responded to the inflammation induced by DSS and bacterial infection..." Except that $Fc\epsilon R1\gamma$ knockout did not affect the DSS model?

Reviewer #3 (Remarks to the Author):

In this manuscript, Huang et al. found that ILC3s highly expressed FcεR1γ gene that encodes Fc-receptor common gamma chain (FcεR1γ) and analyzed the role of FcεR1γ in the expression of surface molecules by using Rorc-cre⁺ Fcer1g^{f/f}. The deficiency of FcεR1γ in ILC3s significantly decreased the surface expression of NKp46 and CD16 and the aspartic acid at position 29 in FcεR1γ was important for the surface expression of NKp46 and CD64.

The authors also analyzed the role of FcεR1γ in the host defense against intragastric infection with *Citrobacter rodentium* and intravenous infection with *Candida albicans* by using Rorc-cre⁺ Fcer1g^{f/f}. Compared to the control mice, Rorc-cre⁺ Fcer1g^{f/f} were more susceptible to infection with these pathogens and the frequency of IL-17A and IL-22 expressing ILC3s in intestine was significantly decreased. The intraperitoneal injection of IL-17A partially restored anti-fungal immunity. Collectively, the authors described that FcεR1γ in ILC3s played an important role in the defense against bacterial and fungal infection and concluded that FcεR1γ functions as a core adaptor protein that stabilizes the expression of its cell membrane partners in ILC3s for mounting an effective anti-infection immunity.

The important role of FcεR1γ in the surface expression of NKp46 and CD16 in ILC3 is interesting and it is nice that the authors found that the aspartic acid at position 29 in FcεR1γ is important. However, I have some concerns regarding the functions of FcεR1γ in ILC3 in the host defense against bacterial and fungal infections. Please see below for the specific comments.

Major comments:

1. The role of FcεR1γ in ILC3 in protection against *Citrobacter rodentium* infection

Compared to the control mice, the frequency of IL-17A and IL-22 expressing ILC3s in intestine was significantly decreased after *C. rodentium* infection (Figure 3). However, the authors did not show that the decreased cytokine production by ILC3 was responsible for the increased bacterial number and more tissue damages. Because the authors described as "the deficiency of FcεR1γ also led to the absence of CD64 in BMDMs (Figure S2A and page 6, line126-127), the deficiency of FcεR1γ may influence on the functions of CD64+ cells (such as macrophages) in intestine. To confirm that FcεR1γ in ILC3 plays an important role in protection against *C. rodentium* infection, it is important to analyze that the functions of CD64+ cells (such as macrophages) in intestine are not affected by the deficiency of FcεR1γ.

Also, the additional experiments would be necessary to clarify that the decreased frequency of IL-17A and IL-22 expressing ILC3s in intestine was responsible for the increase of bacterial number and more severe tissue damages after *C. rodentium* infection.

2. The role of FcεR1γ in intestinal ILC3 in protection against systemic *Candida albicans* infection

The authors described as "ILCs have been suggested to respond to fungal infection" (Page 8, line 176) and cited the reference No. 27. In the referenced study, the authors analyzed ILCs after intragastric infection with *Candida albicans*. However, in this manuscript, the authors intravenously infected mice with *C. albicans* and analyzed ILC3s in intestine. The differences of gene expression and expression of IL-17A and IL-22 in intestinal ILC3s between FcεR1γ deficient mice and control mice are interesting (Figure 5). However, how do intestinal ILC3s respond to *C. albicans* infected intravenously?

Because many other cells except for intestinal ILC3 contribute to host defense against systemic *C. albicans* infection, the functions of other cells in several organs/tissues including spleen and kidneys should be examined in Rorc-cre⁺ Fcer1g^{f/f} mice and control mice. In the same line with mentioned above, the functions of CD64+ cells (such as macrophages) in several organs/tissues including spleen, kidney and intestine should be analyzed at least.

The authors showed that intraperitoneal injection of IL-17A ameliorated the *C. albicans* infection (Figure 6). However, systemic administration of IL-17A stimulate many immune cells in various

organs/tissues. Therefore, this result does not show that IL-17A production by intestinal ILC3 is important for the protection against systemic *C. albicans* infection. The authors should perform other experiments to show the importance of IL-17A production by intestinal ILC3 in the defense against systemic *C. albicans* infection.

3. NKp46 and CD16/FcεR1γ-mediated response in the functions of ILC3s

The authors have shown that the expression of NKp46 and CD16 is diminished or significantly decreased in the deletion of FcεR1γ in ILC3s. However, the mechanism of NKp46 and CD16/FcεR1γ-mediated response in ILC3s has not been well characterized. Do NKp46 and CD16 recognize bacterial or fungal components? To address this question, it may be worth testing whether NKp46/FcεR1γ or CD16/FcεR1γ mediated signaling is stimulated in ILC3s, when these cells are stimulated with *C. rodentium*, *C. albicans* or the sonicates of these microbes. If NKp46/FcεR1γ and CD16/FcεR1γ are not activated by these microbes or microbial components, how these molecules are activated? How these molecules contribute to the defense against bacterial and fungal infections?

It is not necessary to identify the ligands of NKp46 and CD16. However, it would be helpful to strengthen the importance of NKp46 and CD16/FcεR1γ-mediated response in the functions of ILC3s by performing these experiments.

Minor comments

1. Figure S1A

It is hard to distinguish 44 cell populations by the difference of colors. Each cell population in the figure should be labelled with the number.

2. Figure S1C

The information of each population cannot be read, because they are not clearly shown.

Overview

We thank the Reviewers for their interest in our manuscript and thoughtful comments. We are grateful for the specific comments raised, which we addressed in detail in the revised manuscript by additional experiments, analyses, and revision of the text and figures. We first highlight the key revisions and then address each question in a point-by-point manner below.

Key highlights within our revised manuscript include:

1. New data demonstrating that CD16 is the co-receptor of FcεR1γ in supporting ILC3s activation and anti-infection immunity. We generated CD16 and NKp46 KO mice and observed that the loss of CD16, but not NKp46, impaired ILC3 activation in response to pathogen infection.
2. New data further demonstrating the cell-intrinsic function of the IgG-CD16-FcεR1γ pathway in ILC3s. We generated *Rag*^{-/-} *Fcer1g* cKO mice and found that the protective function of FcεR1γ expression in ILC3s is dependent on the presence of IgG antibodies.
3. New data showing the essential residues in FcεR1γ for maintaining CD16 and NKp46 expression in ILC3s.
4. New immunoprecipitation-mass spectrometry (IP-MS) data showing potential binding partners of FcεR1γ and their enriched signaling pathways.
5. Improved clarity and flow of the text.

Reviewers' Comments:

Response to Reviewer #1

*The manuscript by Huang et al. entitled “ Antibody Fc-receptor common gamma chain FcεRIγ stabilizes cell surface receptors in group 3 innate lymphoid cells and promotes anti-infection immunity” seeks to investigate the role of the FcεRIγ chain in ILC3 functions. **ILC3s appear to be able to sense the environment through the integrative reception of different environmental signals. In this context, the analysis of new potential receptors is an important point to evaluate ILC functions. Here, the authors provide interesting observations on the potential role of FcεRIγ+ ILC3s under steady state and inflammatory conditions. However, while the mouse model used seems relevant and well-constructed, as reflected in the beginning of the paper, I am not convinced by the rest of the paper. The key limitations of the study in my opinion are:***

- The lack of clearly identified co-receptor(s) in NKp46+ and CCR6+ ILC3s that can explain the observed phenotypes.

- The lack of a functional mechanism associated with the FcεRIγ in ILC3s.
- The relevance of *in vitro* models used to justify the role of FcεRIγ in NKp46/CD16 expression in ILC3s
- The lack of a complete analysis of the other RORγt+ cells in steady state and inflammatory conditions.
- The relevance of the *C. Albicans* model used to analyze the function of intestinal ILC3s.
- The lack of a convincing discussion.

For these reasons, the paper appears too preliminary to be published in 'Nature Communications'.

However, I encourage the authors to explore this further. Details are provided below.

We are grateful to the reviewer for recognizing the interest and importance of our work and acknowledging the robustness of our mouse model. In response to the reviewer's valuable feedback, we have conducted further experiments to address each of their concerns.

Figure 1

It is unclear whether CD16 is expressed in ILC3 subsets. CCR6+ and CCR6- ILC3s do not appear to express CD16 (Fig.1G), in contrast to the appearance of CD16 expression (Fig.1A/B/C). In addition, CCR6+ do not appear to express Fcgr3 transcripts (Fig.1J). What is the expression of CD16 on CCR6+ ILC3s? The authors need to clarify this point.

We appreciate the reviewer for raising these questions, which we address below:

1. We have re-analyzed CD16 expression on ILC3s using a new antibody (clone# S17014E, BioLegend), and observed that both CCR6+ and CCR6- intestinal ILC3s exhibit high expression of CD16 (**Revised Fig. 1G**). To ensure the specificity of the staining, we included ILC3s pre-blocked with purified Fc-blocking antibody (clone# 2.4G2) as a negative control (**Revised Fig. S1I**). Furthermore, we have observed that the deficiency of FcεRIγ significantly reduces CD16 expression (**Revised Fig. 1G and S1I**). We speculate that the previously observed low expression of CD16 was due to the loss of the antigen binding site for the anti-mouse CD16/32 antibody (clone# 93) during enzymatic tissue digestion.
2. With respect to the presence of *Fcgr3* transcripts in CCR6+ and CCR6- intestinal ILC3s, we have confirmed their existence using bulk RNA-seq. However, CCR6+ ILC3s exhibit lower expression levels compared to CCR6- cells (**For Reviewer Fig. 1 below**).

As for the scRNA-seq data, the lack of robust detection of *Fcgr3* transcripts in all ILC3s can be attributed to the "dropout" effect, where a gene is observed at a low or moderate expression level in one cell but is not detected in another cell of the same cell type. It is a common characteristic/issue of scRNA-seq data (Kharchenko et al., Nat Methods, 2014). Consistently, even cNK cells, a well-known cell type that expresses FcεRIγ, exhibited low detection of *Fcgr3* transcripts in the scRNA-seq dataset (**revised Fig. 1A**).

For Reviewer Fig. 1

If all ILC3s express CD16, what is the role of this receptor? To my knowledge, this role has not been described in the literature, and the authors intend that CD16 contributes to ILC3 function, but how? Furthermore, if CD16, like NKp46, is not expressed on the surface of CCR6⁺ ILC3s, what is/are the co-receptor(s) for FcεRIγ on CCR6⁺ ILC3s? and what is/are their role(s)?

We thank the Reviewer for raising these questions. As mentioned in our response to the first major point, we have confirmed that both CCR6⁺ and CCR6⁻ intestinal ILC3s express CD16. In the revised manuscript, we have further addressed the functionality of CD16, as well as NKp46, by generating new CD16⁻ or NKp46-deficient mouse strains. Our findings demonstrated that the loss of CD16 impairs inflammatory cytokine production in both CCR6⁺ and CCR6⁻ intestinal ILC3s in response to the infection (**Revised Fig. 7A,7B, S8A and S8B**). This provides evidence for the functional role of CD16 in ILC3s.

Aside from CD16, we have performed immunoprecipitation-mass spectrometry (IP-MS) assays to identify other potential binding partners of FcεRIγ. Due to the rarity of ILC3s, we conducted these experiments using the macrophage cell line RAW264.7, given the conserved functions of FcεRIγ in both ILC3s and macrophages. The analyses revealed the binding of JAK1 with FcεRIγ, along with PPP2R1B, PPP2R5A that belong to IL-6 signaling pathway

and ARFGEF1, KPNB1, ARFGEF3, GBF1, CYFIP1, ELMO1, DOCK2, HACD3 which are associated with small GTPase signaling pathway (**Revised Fig. S7**). These findings suggested that FcεR1γ is involved in activation cascade transduction from cytoplasmic to nucleus. However, further research is necessary to fully understand the function of these pathways facilitated by FcεR1γ for pathogen sensing and anti-infection immunity. We have discussed this point in the revised **Discussion** (p.14).

CD16 is the type III Fcγ receptor that binds the Fc portion of IgG antibodies. Did the authors test this feature? Did the authors test the functions of FcεR1γ+ ILC3s in the absence of adaptive cells? Knowing that ILC3s have a more important functionality in Rag1/2-/- mice, and thus in the absence of B cells and IgG, what would be the role of CD16 in this case?

We appreciate the Reviewer for these important suggestions. To investigate the function of FcεR1γ in ILC3s in the absence of adaptive cells, we generated *Rag1*^{-/-} CKO (*Rag1*^{-/-}*Rorc-cre*⁺*Fcer1g*^{fl/fl}) mice and control (*Rag1*^{-/-}*Rorc-cre*⁻*Fcer1g*^{fl/fl}) mice. Interestingly, we found that *Rag1*^{-/-} CKO mice show no significant difference in the fatality induced by fungal infection compared to control mice (**Revised Fig. 7C**). These results suggest that components of the adaptive immune system are essential for the FcεR1γ-mediated anti-infection functions of ILC3s. Importantly, the replenishment of mouse IgG into control mice, but not *Rag1*^{-/-} CKO mice, significantly enhanced the host anti-fungal immunity (**Revised Fig. 7D-F, S8C and S8D**). These findings demonstrated that FcεR1γ-mediated anti-infection immunity is dependent on the presence of IgG.

In addition, we performed *in vitro* binding assays to investigate whether ILC3s could be bound by different IgG isotypes. While IgG antibodies showed robust binding to the surface of RAW cells, we did not observe binding of IgG on ILC3s (**For Reviewer Fig. 2**). This discrepancy may be due to the fact that low-to-medium affinity Fc receptors, including CD16, can bind IgG only when it is present in an immune complex, aggregated, or opsonized by other components¹. On the other hand, high-affinity Fc receptors, such as CD64 on macrophages, can bind free/monomeric IgG. This observation might explain why CD16 deficiency caused by FcεR1γ knock-out leads to a cytokine production defect during infection but not at homeostasis. We have discussed this part in the revised **Discussion** (p.14).

For Reviewer Fig. 2

Minor comment: The figure 1B needs to be revised. The ILC3 subsets are not clearly visible (CCR6^{+/-}) and the control isotype is missing (bottom panel).

We have revised the panel according to the suggestion (**Revised Fig. 1B**).

Figure 2

Figure 2A is misleading. It may give the impression that ILC3s were used in this experiment, but a HEK model was used. This model appears to be useful for studying the importance of certain residues, but not for verifying the maintenance of receptors in ILC3s. To fully understand their role in Nkp46 and CD16 expression, it would be more appropriate and comprehensive to study ILC3s or a similar model.

We apologize for the confusion. We have revised the panel to clearly indicate that BMDMs were utilized for alanine scanning mutagenesis analysis (**Revised Fig. 2A**). Additionally, to validate the function of D29 and L39 residues in FcεR1γ in maintaining NKP46 and CD16 expression in ILC3s, we isolated intestinal ILC3s for *in vitro* expansion and transfection with lentivirus expressing WT FcεR1γ, FcεR1γ^{D29A} or FcεR1γ^{L39A} (**Revised Fig. S2C-E**). Although the overall transfection efficiency in ILC3s was low (**Revised Fig. S2F**), the expression of exogenous WT FcεR1γ enabled the detection of a subset of NKp46⁺ cells in CCR6⁻ ILC3s (**Revised Fig. 2D**). However, the expression of FcεR1γ^{D29A} or FcεR1γ^{L39A} failed to induce the expression of NKp46 (**Revised Fig. 2D and 2E**). Furthermore, we found that WT FcεR1γ, but not the FcεR1γ^{D29A} or FcεR1γ^{L39A}, increased the expression of CD16 in both CCR6⁻ and CCR6⁺ ILC3s (**Revised Fig. S2G,2H**). These findings demonstrated that the aspartic acid at position 29 and leucine at position 39 are essential and conserved amino acids in FcεR1γ for maintaining the abundance of its binding partners, including NKp46 and CD16 in ILC3s, as well as CD64 in macrophages.

Figure3

The authors observed a very low percentage of IL-22 producing cells after stimulation. (Figure 3F-I). From different publications (and different labs), it appears that the percentage of IL-22 in ILC3s is around 25-40% (Zhou et al. Nat. Immunol. 2022; Giacomini et al. JEM 2019 ; Talbot et al. Nature 2020). Therefore, stimulation does not seem to be appropriate or effective in these experiments. The low percentage of IL-17⁺ ILC3s and the lack of differences between CCR6⁺ and CCR6⁻ ILC3s are also 'suspicious' for the optimal cytokine detection. Further experiments are necessary as low cytokine detection may interfere with the interpretation of the results. Indeed, I note that with optimal cytokine detection, the authors did not observe any differences between WT and KO mice (Figure S3E-G).

We appreciate the reviewer for the comment. The intracellular cytokine abundance can be influenced by various factors, including the tissue source (such as duodenum, jejunum, ileum, colon), infection model (bacteria or fungi), animal facility (microbiota), and the stimulation protocol used *ex vivo*. The production of IL-22 from intestinal ILC3s is even to be orchestrated by circadian rhythm and food intake^{2,3}. However, there is currently no widely accepted standard protocol for stimulation.

In the study mentioned by the reviewer, Wang et al. observed approximately 7% of IL-22⁺ ILC3s from siLPs after stimulation with IL-23 (10 ng/ml) for 4 hours, with Golgi Plug added for the last 3 hours (Fig. 3E)⁴. In Zhou's work, IL-23, IL-1 β , and TNF- α stimulation for 4 hours resulted in 10-20% IL-22⁺ sort-purified siLP ILC3s and approximately 30% IL-22⁺ cLP ILC3s (Fig. S6b and Fig. S5e from Zhou et al. Nat. Immunol. 2022)⁵. Another publication by Zhong et al. reported that stimulation with IL-7 and IL-23 for 3 hours led to 10-40% IL-22⁺ siLP ILC3s and 20-60% IL-22⁺ cLP ILC3s (Fig. 6d-f from Zhong et al. Nat. Immunol. 2016)⁶. Seillet et al. demonstrated that stimulation with PMA and ionomycin for 4.5 hours resulted in 10-15% IL-22⁺ sort-purified siLP ILC3s (Fig. 1A, from Seillet et al. Nat. Immunol. 2019)³. In Talbot et al.'s publication, 25% IL-22⁺ sort-purified ileum ILC3s were observed after 16 hours of culture with IL-23, followed by 4 hours of stimulation with PMA, ionomycin, and Golgi Plug (Fig. 2, Talbot et al. Nature 2020)². Moreover, Luccia et al. stimulated siLP ILC3s with IL-23 and IL-1 β for 6 hours in a *C. rodentium*-infected mouse model, resulting in a 20-30% IL-22⁺ population (Fig. 2G, Luccia et al. JEM, 2019)⁷. It is important to note that the stimulation protocols (duration, cytokines, cell source) largely depend on the specific perturbation the researcher aims to apply to the cells ex vivo. Therefore, some previous studies have also analyzed cytokine production without stimulation (Fig. 4D, Huang et al. Immunity, 2021)⁸. The production of IL-17A shows a similar pattern to IL-22 in ILC3s across all these publications. In conclusion, the levels of cytokine expression are predominantly determined by the duration of artificial stimulation with nonspecific stimulants (such as PMA and ionomycin). However, this process does not significantly alter the overall differences among the experimental groups.

In our study, we aimed to explore the cell function more related to the intrinsic state of ILC3s. Instead of the regular 4-16 hour stimulation with pro-inflammatory cytokine cocktails (IL-23, IL-1 β , and/or TNF- α), we chose to stimulate the sorted purified siLPL for only 2.5 hours in the presence of IL-23. Therefore, it is reasonable that we detected approximately 10-20% IL-22⁺ in siLP ILC3s (**Fig. 3F-G**, *C. rodentium* infected) and 20-40% IL-22⁺ in cLP ILC3s (**Fig. S3F**, DSS-induced colitis). Interestingly, in mice infected with *C. albicans*, we observed reduced cytokine production compared to *C. rodentium*-infected or naïve mice. Furthermore, Fc ϵ r1g-CKO ILC3s showed lower cytokine production compared to control ILC3s after *C. albicans* infection, and the difference in mRNA levels supports the conclusion of intracellular cytokine staining (**Fig. S5G**).

In Figures S3E-G, to determine the function of ILC3s in DSS-induced colitis, we used the same stimulation protocol for ILC3s isolated from the cLP (but not siLP) and observed 20-40% IL-22⁺ in cLP ILC3s.

In response to the reviewer's concern, we conducted further experiments to determine intracellular cytokine expression with a 4-hour stimulation. Under this protocol, we observed 20-40% IL-22⁺ and 10% IL-17A⁺ in WT ILC3s from siLPs (**Revised Fig. 7A-F and S8**).

How do the authors explain the phenotype of RorcCre⁺FcεRIγ mice with C. Rodentium? How can the FcεRIγ receptor cause this defect in cytokine production?

We thank the reviewer for these questions. In the revised manuscript, we have demonstrated that FcεRIγ deficiency resulted in the loss of CD16, which promoted inflammatory cytokine production in ILC3s (**Revised Fig. 7A,B**). Moreover, we revealed the crucial function of IgG in FcεRIγ-mediated cytokine production and anti-infection immunity (**Revised Fig. 7C-F**). Additionally, we have explored potential signaling pathways mediated FcεRIγ using IP-MS analysis (**Revised Fig. S7**). These findings provide insights into the molecular mechanisms underlying FcεRIγ-mediated functions in ILC3s in response to the infection. We kindly direct the reviewer to the detailed response to these questions in our response to questions #2 and #3 related to Figure 1 above.

The authors do not describe an associated receptor in CCR6⁺ ILC3s. So how can FcεRIγ block the production or secretion of IL-22 and IL-17? Similarly, what is the mechanism in NKp46⁺ ILC3s? The authors can argue a role for NKp46. However, Wang et al. (Plos Biol 2018) have already investigated the potential role of NKp46. Using an NKp46 deficiency model (Ncr1gfp/gfp mice), they demonstrated that NKp46 is required for ILC1 development, but does not affect the ILC3 development and their IL-22 production. Satoh-Takayama et al. showed also that NKp46 is not required for IL-22-mediated ILC3 in the gut to defend against C. Rodentium (J. Immunol 2009), so how can the authors explain the observed phenotype in the absence of a clearly identified co-receptor? Can the authors explain the difference in phenotype of ILC3s with CR and DSS? Is the FcεRIγ not 'engaged' in the DSS model?

We appreciate the reviewer's questions and comments. In the revised manuscript, we have provided evidence to demonstrate that the loss of CD16, but not NKp46, significantly impairs IL-17 production in total ILC3s (**Revised Fig. 7A,B**). Therefore, we propose that CD16 serves as a functional co-receptor of FcεRIγ on ILC3s for anti-infection immunity. We acknowledge that the differential requirement for CD16-FcεRIγ complex expression on ILC3s in the context of healthy and DSS-treated conditions versus infection conditions is an interesting topic for further investigation. However, as it falls outside the scope of this study, we have not delved into this aspect in the revised manuscript. Nonetheless, we recognize the importance of exploring the contextual regulation of CD16-FcεRIγ signaling in ILC3s in different physiological and pathological conditions, and it could be a subject of future research. We have discussed this part in the revised **Discussion** (p.14).

This study shows that T cells are not affected in RorcCre+Fcer1f/f mice under steady-state conditions. However, the effect of the Fcer1g deletion on inflammatory T cells in the DSS, C. Rodentium and C. Albicans models remains unknown. The authors cannot exclude an expression of FcεRIγ in inflammatory RORγt+ T cells or IELs, which may influence the inflammatory response and the development of infection. In a recent study, Chou and colleagues defined a distinct population of innate like T cells with cytotoxic potential, called αβILCTCKs, that exhibit reactivity to tumor antigens (Chou et al. Nature 2022). They demonstrated that this population is replenished by thymic progenitors and infiltrates the tumor such as pancreatic cancer or colorectal cancer tissue. We already know that the T cell population is composed of a large diversity of resident, activated and memory cells that are sensitive to the environmental cues and are plastic (Ribot et al Nat. Rev. Immunol. 2021; McDonald et al. Nat. Rev. Immunol. 2018; Caruso et al. Nat. Rev. Immunol. 2020). As observed in the tumor model, FcεRIγ+ RORγt+ T cells could appear during C. Rodentium (or C. Albicans) infection and affect the intestinal barrier. Due to the crucial role of the T cell response during the C. rodentium infection (Stockinger Curr. Opin. Microbiol. 2021), the role of RORγt+ T cells in RorcCre+Fcer1f/f mice should be addressed.

We appreciate the reviewer's comments and suggestions. We are aware of the newly defined population of innate-like T cells (αβILCTCKs) in tumors that express FcεRIγ (Chou et al., Nature 2022). However, in our study, we did not observe co-expression of FcεRIγ and RORγt in T cells using flow cytometry analysis (**Revised Fig. 1C**), which is consistent with our scRNA-seq dataset (**For Reviewer Fig. 3**). Additionally, in the revised manuscript, we

confirmed that there were no FcεRIγ⁺RORγt⁺ cells in the T cell or myeloid cell populations in mice infected with *C. rodentium* or *C. albicans* (**Revised Fig. S6D and S6E**). Furthermore, we demonstrated that the protective immunity mediated by FcεRIγ expression on RORγt⁺ cells was lost in *Rag1*^{-/-} mice but could be rescued by IgG transfer. This highlights the importance of the IgG-CD16-FcεRIγ pathway and its cell-intrinsic function in ILC3s in response to infection.

For Reviewer Fig. 3

Minor comment: In Fig.3 F-I, the authors should indicate the organ used for these experiments (SI or Colon?)

We have revised these panels according to the suggestion (**Revised Fig. 3F-I**).

Figure 4/5

The authors chose to administer C.Albicans by intravenous injection (Fig4.A). This choice seems somewhat surprising given that the analysis focuses on intestinal ILC3s and not peripheral ILC3s. Did the authors verify the presence of C. Albicans in the gut? Furthermore, the authors focused on pathological signs in the kidney (Fig.4 C-F), but what is the phenotype and activation of ILC3s in this organ? As mentioned by the authors, ILC3 may be a source of TH17 cytokines during the early stages of infection (Gladiator et al. J Immunol 2013). However,

another group, using *Il17a* reporter mice, reported that *TH17* and $\gamma\delta$ T cells are the main source of *IL-17* (Ramani et al. *JCI* 2018). While *ILC3s* (with redundant function with T cells) have been described in the infected tongue (by oral gavage; Break et al. *Science* 2021), *Il17YFP+ ILC3* were not detected in either sham or *C. Albicans*-infected kidneys (Ramani et al. *JCI* 2018).

Based on this route of transmission, it can be assumed that *C. Albicans* is able to infect all organs independently of the intestine. Therefore, intestinal *ILC3s* alone would be able to interfere with the infection in the periphery? How could this happen?

We thank the reviewers for these questions, which we address below:

1. The essential function of *ILCs* in response to *C. albicans* infection in the tongue has been highlighted in the study using *Rorc*^{-/-} and *Rag*^{-/-} mice⁹. Additionally, *IL-17* secreted from *ILCs* has been shown to play an important role in the sublingual infection of *C. albicans*^{9,10}. *C. albicans* is typically a harmless commensal organism but can be an opportunistic pathogen for immunologically weak and immunocompromised people¹¹. *C. albicans* is unable to colonize and infect the intestines of wild-type mice unless pre-treated with antibiotics or *DSS*^{12,13}, both of which can perturb *ILC3* development and activation. Therefore, in our study, we employed a commonly used systemic infection model through intravenous injection of *C. albicans*¹⁴ to study the response *ILC3s* (Fig. 4A). Importantly, *ILC3s* express various cytokine receptors that can respond to systemic infection-induced inflammatory environmental changes¹⁵. We confirmed that intestinal *ILC3s* responded to systemic infection (Fig. S4). Moreover, systemic *C. albicans* infection induced severe mortality, which allows us to directly determine the protective immunity conferred by *FcεR1γ* expression in *ILC3s*.
2. Our data clearly demonstrated that intestinal *ILC3s* responded to the systemic infection (Fig. S4). The systemic infection causes multi-organ damage, including the intestine, liver, spleen, lung, and brain. however, fungal hypha is mainly present in the kidney. So the CFU in the kidney is an indicator of the severity of candidiasis. Similarly to previous findings that showed no *ILC3s* present in the site of oral *C. albicans* colonization (the tongue), we observed that no *RORγt*-expressing *ILC3s* were detected in the kidney (For Reviewer Fig. 4).

3. In the revised manuscript, we further showed that ILC3s, but not other immune cell types, reduced the expression of IL-17A (**Revised Fig. S6C**). We also confirmed the systemic reduction of IL-17A in the serum of CKO mice (**Revised Fig. 6F**). Together, these data indicated that intestinal ILC3s contributed to IL-17A levels in the periphery for host defense against infection. We acknowledge that IL-17A from other cells, such as TH17, $\gamma\delta$ T, and NK cells, also contributes to host defense against *C. albicans* infection. Through data obtained from CKO and Rag^{-/-} CKO mice with IgG transfer, our work highlights the important and non-redundant role of IL-17A from ILC3s in antifungal immunity.

For Reviewer Fig. 4

The authors studied the transcriptional state of ILC3s in the gut after infection, but the lack of analysis of RorcCre-Fcer1ff and RorcCre+Fcer1ff ILC3s in steady-state conditions is detrimental to interpretation (Fig5A, S4E, S5C, S5E).

We analyzed the transcriptional state of CKO and control ILCs via scRNA-seq at the steady state and failed to observe significant changes in the proportions (**Revised Fig. 11**) and transcriptional states (**Revised Fig. S1L**). Additionally, we performed bulk RNA-seq analysis for ILC3s at the steady state and obtained similar results. To avoid repetition, we did not include these data in the manuscript.

As mentioned above, TH17 cells have an important role in the host protection, what is the phenotype and function of intestinal and renal T cells in C. Albicans infected RorcCre-Fcer1ff and RorcCre+Fcer1ff mice?

As noted above, the production of IL-22 and IL-17 does not seem to be in agreement with observations from other labs (Fig5D-F). It seems necessary to re-evaluate this part of the experiment in order to best interpret the results.

As mentioned earlier, we did not observe co-expression of FcεRIγ and RORγt in T cells in the intestines of naïve mice (**Revised Fig. 1C**) or mice infected with *C. rodentium* or *C. albicans* (**Revised Fig. S6D and S6E**). Furthermore, we demonstrated that the protective immunity mediated by FcεRIγ expression on RORγt⁺ cells was lost in *Rag1*^{-/-} mice but could be rescued by IgG transfer. This highlights the importance of the IgG-CD16-FcεRIγ pathway and its cell-intrinsic function in ILC3s in response to infection. We have carefully considered the reviewer's suggestions and have made significant revisions to the manuscript accordingly. We believe that these revisions have addressed all the questions and concerns raised by the reviewer, making our conclusion more convincing.

Response to Reviewer #2

*The manuscript “Antibody Fc-receptor common gamma chain 1 FcεRIγ stabilizes cell surface receptors in group 3 innate lymphoid cells and promotes anti-infection immunity” by Huang et al described the role of the common Fc epsilon receptor gamma chain in a select mucosal immune cell group. **In general, the manuscript is well written and the data generally compelling.** It is not surprising that removing this protein from a cell limits surface expression of Fc gamma receptor 1 and 3a (in humans) and other receptors; this is well known. Thus, the novelty lies not in the reduced surface expression resulting from the knockout, but rather in the impacts associated with these knockout. The authors demonstrate that a common colon damage model using DSS is not affected by the knockout, but responses to bacterial and fungal pathogens are. A major objection to the conclusions drawn results from the use of the word “regulates.” FcεRIγ cannot be seen as a regulator unless its expression changes naturally, and thus imparts a change on other interacting proteins. The authors present no evidence supporting this assertion. Rather, FcεRIγ is necessary for the surface expression of select receptors that mediate select responses. All occurrences of interpretations relating to regulation should therefore be revised.*

We appreciate the reviewer's recognition of the compelling nature of our data. We completely agree with the reviewer's comment regarding the use of the word "regulates". We have carefully revised the text to adjust the interpretations related to the regulation accordingly. Furthermore,

we have included new data to enhance the novelty of our findings and to show potential co-receptor(s) on the surface of ILC3s that are responsible for the observed phenotypes.

A few other comments:

Do ILC3s express CD3zeta, which can also substitute for FcεR1γ to promote CD16a surface localization? Low expression may explain the residual protein levels in some of the knockouts (Figure 1G, for example).

Based on our bulk RNA-seq data, we observed that ILC3s barely express CD3zeta (encoded by *Cd247* gene, TPM<5). Furthermore, we did not observe any significant changes in *Cd247* expression in FcεR1γ CKO ILC3s in response to *C. albicans* infection (**For Reviewer Fig. 5**).

For Reviewer Fig. 5

Figure 4D. it is unclear how the sites of fungal invasion are chosen, they appear randomly chosen to this naïve reviewer.

We apologize for not providing an explanation earlier regarding our choice of the systemic fungal infection model. The rationale behind selecting this model is based on previous studies that have indicated the involvement of innate lymphoid cells (ILCs) in defense against sublingual *Candida albicans* infection⁹. *C. albicans* is normally a harmless commensal organism but can be an opportunistic pathogen for immunologically weak and immunocompromised people¹¹. *C. albicans* is unable to colonize and infect the intestines of wild-type mice unless pre-treated with antibiotics or DSS^{12,13}, both of which can perturb ILC3 development and activation. Therefore, we employed a systemic infection model through intravenous injection of *C. albicans*¹⁴ to study the response of ILC3s. Moreover, systemic *C. albicans* infection induced severe mortality, which allows us to directly determine the protective immunity conferred by FcεR1γ expression in ILC3s. We have revised the text to

provide a clearer explanation of the rationale for using the systemic fungal infection model in the revised manuscript.

Line 98, it would be helpful to describe the justification for choosing the Rorc promoter to drive cre expression, and introduce the caveats described later.

We thank the reviewer for the suggestion. Rorc-cre strain is the most widely used genetic tool that targets ILC3s. In the revised manuscript, we have included a demonstration of the specific deletion of FcεR1γ in RORγt-expressing ILC3s in CKO mice as compared to control mice (**Revised Fig. 1C**). Consistently, our RNAseq data analysis revealed that CD3⁺ T cells, including RORγt⁺ Th17, Treg cells and γδT cells that are also targeted by the Rorc-cre, did not express FcεR1γ in conjunction with RORγt (**For Reviewer Fig. 3 above**). Furthermore, we provided new data showing that RORγt⁺ T cells still did not express FcεR1γ in mice infected with *C. rodentium* or *C. albicans* (**Revised Fig. S6D and S6E**). We have revised the text and figures accordingly.

Line 109. “Completely diminished” is an odd phrase. Eliminated or significantly reduced would be more precise.

We have revised the text accordingly.

The alanine scanning experiments using BMDM are only tangentially related to the core of the paper, that FcεR1γ knockout affects multiple functions. The identification of D29 doesn't impact the more impactful experiments, including the bacterial and fungal infection models. This section could be removed to improve conciseness.

We appreciate the reviewer's suggestion. In the revised manuscript, we have conducted experiments to validate the functional importance of these residues in maintaining CD16 and NKp46 expression in ILC3s (**Revised Fig. 2E, 2F, Fig. S2D-S2H**). While we agree with the reviewer that it would be ideal to further investigate the roles of these residues in anti-infection immunity *in vivo*, we regret to say that it is currently not feasible to perform such experiments. This is primarily due to the unavailability of a mouse strain containing these mutations in FcεR1γ.

Line 155. “In consistent,” is unclear

We apologize for the confusion. We have revised the sentence as “Consistent with previous studies that demonstrated the importance of cytokine production, especially IL-22 and IL-17A, from intestinal ILC3s in defense against *C. rodentium*¹⁶, we found that the frequencies of both IL-22- and IL-17A-expressing ILC3s were significantly decreased in CKO mice compared to control mice (**Revised Figures 3F-3I, and S3L**).” (p.8).

Line 173. “The distinct requirement of *FcεR1γ* in ILC3s responded to the inflammation induced by DSS and bacterial infection...” Except that *FcεR1γ* knockout did not affect the DSS model?

We thank the reviewer for the question. We have shown that the loss of *FcεR1γ* in ILC3s did not alter the colon length (**Fig. S3C**), intestinal histology score (**Fig. S3D**), frequencies of total ILC3s (**Fig. S3E**) or IL-22 or IL-17A-expressing ILC3s (**Fig. S3F and S3G**). These results demonstrate that *FcεR1γ* does not play a critical role in ILC3s in response to DSS-induced inflammation. In contrast, we observed that *FcεR1γ* deficiency in ILC3s significantly increased the bacterial loads and decreased frequencies of both IL-22- and IL-17A-expressing ILC3s in the intestines in response to *C. rodentium* infection (**Fig. 3F-3I, and S3L**). Thus, *FcεR1γ* supported ILC3-mediated host defense immunity against bacterial infection. We have revised the sentence as “Given that *FcεR1γ* expression in ILC3s participated in the inflammation induced by the bacterial infection but not DSS treatment, we further investigated *FcεR1γ* function in ILC3s in response to other pathogens.”

Response to Reviewer #3:

*In this manuscript, Huang et al. found that ILC3s highly expressed *Fcer1g* gene that encodes Fc-receptor common gamma chain (*FcεR1γ*) and analyzed the role of *FcεR1γ* in the expression of surface molecules by using *Rorc-cre⁺ Fcer1g^{fl/fl}*. The deficiency of *FcεR1γ* in ILC3s significantly decreased the surface expression of NKp46 and CD16 and the aspartic acid at position 29 in *FcεR1γ* was important for the surface expression of NKp46 and CD64.*

*The authors also analyzed the role of *FcεR1γ* in the host defense against intragastric infection with *Citrobacter rodentium* and intravenous infection with *Candida albicans* by using *Rorc-cre⁺ Fcer1g^{fl/fl}*. Compared to the control mice, *Rorc-cre⁺ Fcer1g^{fl/fl}* were more susceptible to*

infection with these pathogens and the frequency of IL-17A and IL-22 expressing ILC3s in intestine was significantly decreased. The intraperitoneal injection of IL-17A partially restored anti-fungal immunity. Collectively, the authors described that FcεR1γ in ILC3s played an important role in the defense against bacterial and fungal infection and concluded that FcεR1γ functions as a core adaptor protein that stabilizes the expression of its cell membrane partners in ILC3s for mounting an effective anti-infection immunity.

The important role of FcεR1γ in the surface expression of NKp46 and CD16 in ILC3 is interesting and it is nice that the authors found that the aspartic acid at position 29 in FcεR1γ is important. However, I have some concerns regarding the functions of FcεR1γ in ILC3 in the host defense against bacterial and fungal infections. Please see below for the specific comments.

We thank the reviewer for appreciating the interest and importance of our work. We have addressed the Reviewer's concerns in detail in the revised manuscript with new experiments to substantiate key points.

Major comments:

1. The role of FcεR1γ in ILC3 in protection against Citrobacter rodentium infection

Compared to the control mice, the frequency of IL-17A and IL-22 expressing ILC3s in intestine was significantly decreased after C. rodentium infection (Figure 3). However, the authors did not show that the decreased cytokine production by ILC3 was responsible for the increased bacterial number and more tissue damages. Because the authors described as "the deficiency of FcεR1γ also led to the absence of CD64 in BMDMs (Figure S2A and page 6, line126-127), the deficiency of FcεR1γ may influence on the functions of CD64+ cells (such as macrophages) in intestine. To confirm that FcεR1γ in ILC3 plays an important role in protection against C. rodentium infection, it is important to analyze that the functions of CD64+ cells (such as macrophages) in intestine are not affected by the deficiency of FcεR1γ.

Also, the additional experiments would be necessary to clarify that the decreased frequency of IL-17A and IL-22 expressing ILC3s in intestine was responsible for the increase of bacterial number and more severe tissue damages after C. rodentium infection.

We thank the reviewer for the comment. The reduction of CD64 in BMDMs was observed in cells isolated from *Fcer1g* germline KO mice. To investigate the cell-intrinsic function of FcεR1γ in ILC3s, we generated *Fcer1g*-conditional knockout (*Rorc-cre⁺Fcer1g^{fl/fl}*, CKO) mice,

specifically targeting FcεR1γ expression in RORγt-expressing cells. In the revised manuscript, we demonstrated that CD11b⁺ cells barely express RORγt in homeostasis (**Revised Fig.1C and For Reviewer Fig. 3 above**), or under inflammatory conditions induced by *C. rodentium* or *C. albicans* infection (**Revised Fig. S6D and S6E**). Furthermore, we also confirmed that the deletion of FcεR1γ in RORγt-expressing cells did not change the expression of CD64 in macrophages in the intestines of CKO mice (**For Reviewer Fig. 6**).

ILC3s have indeed been recognized as crucial players in host defense against intestinal *C. rodentium* infection, primarily through their secretion of IL-22¹⁶. Upon infection, IL-22 from ILC3s activates intestinal epithelial cells to release antimicrobial peptides and mucus via the IL-22R-JAK1-STAT signal transduction pathway, contributing to pathogen clearance. Since this immunological cascade has been extensively described in previous studies, we did not further explore this pathway in our current study. We have revised the text accordingly to introduce this key background information (p.8).

For Reviewer Fig. 6

2. The role of FcεR1γ in intestinal ILC3 in protection against systemic *Candida albicans* infection

The authors described as “ILCs have been suggested to respond to fungal infection” (Page 8, line 176) and cited the reference No. 27. In the referenced study, the authors analyzed ILCs after intragastric infection with *Candida albicans*. However, in this manuscript, the authors intravenously infected mice with *C. albicans* and analyzed ILC3s in intestine. The differences of gene expression and expression of IL-17A and IL-22 in intestinal ILC3s between FcεR1γ deficient mice and control mice are interesting (Figure 5). However, how do intestinal ILC3s respond to *C. albicans* infected intravenously?

Because many other cells except for intestinal ILC3 contribute to host defense against systemic C. albicans infection, the functions of other cells in several organs/tissues including spleen and kidneys should be examined in Rorc-cre⁺ Fcer1g^{fl/fl} mice and control mice. In the same line with mentioned above, the functions of CD64⁺ cells (such as macrophages) in several organs/tissues including spleen, kidney and intestine should be analyzed at least.

The authors showed that intraperitoneal injection of IL-17A ameliorated the C. albicans infection (Figure 6). However, systemic administration of IL-17A stimulate many immune cells in various organs/tissues. Therefore, this result does not show that IL-17A production by intestinal ILC3 is important for the protection against systemic C. albicans infection. The authors should perform other experiments to show the importance of IL-17A production by intestinal ILC3 in the defense against systemic C. albicans infection.

We would like to express our gratitude to the reviewer for their valuable questions, which we have addressed below:

1. Previous studies utilizing *Rorc*^{-/-} and *Rag*^{-/-} mice have demonstrated that ILCs contribute to defense against sublingual *C. albicans* infection⁹. Additionally, another study showed that IL-25 induces inflammatory ILC2s (iILC2s) in the lung, which contribute to antifungal immunity in the tongue¹⁰. Our observation of the absence of RORγt-expressing ILC3s in the tongue is consistent with these findings. As a microorganism commonly found in human barriers, such as the intestine, oral cavity, and vaginal, *C. albicans* can be an opportunistic pathogen for immunologically weak and immunocompromised people¹¹. *C. albicans* is unable to colonize and infect the intestines of wild-type mice unless pre-treated with antibiotics or DSS^{12,13}, both of which can perturb ILC3 development and activation. Thus, there is no optimal protocol for specifically colonizing *C. albicans* in the intestine to study intestinal ILC3 responses. In our study, we employed a commonly used systemic infection model through intravenous injection of *C. albicans*¹⁴ to study the response of ILC3s. Importantly, ILC3s express various cytokine receptors that enable them to respond to systemic infection-induced changes in the inflammatory environment¹⁵. Furthermore, systemic *C. albicans* infection induced severe mortality, allowing us to directly determine the protective immunity conferred by FcεR1γ expression in ILC3s (**Revised Fig. 6A,B**).

- Our data clearly demonstrated that intestinal ILC3s responded to the systemic infection (**Fig. S4**). The systemic infection causes multi-organ damage, including the intestine, liver, spleen, lung, and brain. However, fungal hypha is mainly present in the kidney. So CFU in the kidney serves as an indicator of the severity of candidiasis. Similar to previous findings showing the absence of ILC3s in the site of oral *C. albicans* colonization (tongue), we observed no ROR γ t-expressing ILC3s in the kidney (**For Reviewer Fig. 4**).
- In the revised manuscript, we showed that intestinal ILC3s, but not other immune cell types, exhibited reduced expression of IL-17A (**Revised Fig. S6C**). We also confirmed the systemic reduction of IL-17A in the serum of CKO mice (**Revised Fig. 6F**). While we acknowledge that many other cell types, besides intestinal ILC3s, contribute to host defense against systemic *C. albicans* infection, our CKO mice model did not show significant differences in other immune cell types in the intestines, including macrophages (Macro.), eosinophils (Eos.), neutrophils (Neu.), and inflammatory monocytes (iMono.) (**For Reviewer Fig. 7**). Together, these data indicated that intestinal ILC3s contributed to IL-17A levels in the periphery for host defense against infection. We recognize that other cell types also contribute to host defense against *C. albicans* infection. Nevertheless, our work based on CKO mice emphasizes the important and non-redundant role of IL-17A from ILC3s in antifungal immunity.

For Reviewer Fig. 7

3. NKp46 and CD16/FcεR1γ-mediated response in the functions of ILC3s

*The authors have shown that the expression of NKp46 and CD16 is diminished or significantly decreased in the deletion of FcεR1γ in ILC3s. However, the mechanism of NKp46 and CD16/FcεR1γ-mediated response in ILC3s has not been well characterized. Do NKp46 and CD16 recognize bacterial or fungal components? To address this question, it may worth testing whether NKp46/FcεR1γ or CD16/FcεR1γ mediated signaling is stimulated in ILC3s, when these cells are stimulated with *C. rodentium*, *C. albicans* or the sonicates of these microbes. If NKp46/FcεR1γ and CD16/FcεR1γ are not activated by these microbes or microbial components, how these molecules are activated? How these molecules contribute to the defense against bacterial and fungal infections?*

It is not necessary to identify the ligands of NKp46 and CD16. However, it would be helpful to strengthen the importance of NKp46 and CD16/FcεR1γ-mediated response in the functions of ILC3s by performing these experiments.

We appreciate the reviewer for these important questions and suggestions. In the revised manuscript, we generated CD16 KO and NKp46 KO mice to determine which one was essential for the activation of ILC3s during the infection. We found that both CCR6⁻ and CCR6⁺ ILC3s showed reduced production of IL-17A and IL-22 in the absence of CD16, but not NKp46, when challenged with fungi infection (**Revised Fig. 7A,7B,S8A and S8B**). These data indicate the CD16-FcεR1γ complex supports ILC3 activation and cytokine production. Additionally, we generated *Rag1*^{-/-} CKO (*Rag1*^{-/-}*Rorc-cre*⁺*Fcer1g*^{fl/fl}) mice and control (*Rag1*^{-/-}*Rorc-cre*⁻*Fcer1g*^{fl/fl}) mice. Interestingly, we found that *Rag1*^{-/-} CKO mice show no significant difference in the fatality induced by fungal infection compared to control mice (**Revised Fig. 7C**). However, the replenishment of mouse IgG into control mice, but not *Rag1*^{-/-} CKO mice, significantly enhanced the host anti-fungal immunity (**Revised Fig. 7D-F, S8C and S8D**). These findings demonstrated that FcεR1γ-mediated anti-infection immunity is dependent on the presence of IgG, which is the ligand of CD16.

To explore FcεR1γ-mediated signaling pathways, we have performed immunoprecipitation-mass spectrometry (IP-MS) assays to identify its binding partners. Due to the rarity of ILC3s, we conducted these experiments using the macrophage cell line RAW264.7, given the conserved functions of FcεR1γ in both ILC3s and macrophages. The analyses revealed the binding of JAK1 with FcεR1γ, along with PPP2R1B, PPP2R5A that belong to IL-6 signaling

pathway and ARFGEF1, KPNB1, ARFGEF3, GBF1, CYFIP1, ELMO1, DOCK2, HACD3 which are associated with small GTPase signaling pathway (**Revised Fig. S7**). These findings suggested that FcεR1γ is involved in activation cascade transduction from cytoplasmic to nucleus. However, further research is necessary to fully understand the function of these pathways facilitated by FcεR1γ for pathogen sensing and anti-infection immunity. We have discussed this point in the revised **Discussion** (p.14).

We have tried to directly stimulate ILC3s *in vitro* with *C. rodentium* or *C. albicans*, but did not observe notable activation and cytokine production of ILC3s (data not shown). We propose that, instead of directly sensing these pathogens, ILC3s express receptors, such as CD16-FcεR1γ, to indirectly respond to microenvironmental changes induced by the infection, including proinflammatory cytokines and antibodies.

Minor comments

1. Figure S1A

It is hard to distinguish 44 cell populations by the difference of colors. Each cell population in the figure should be labelled with the number.

We have revised the figure accordingly.

2. Figure S1C

The information of each population cannot be read, because they are not clearly shown.

We apologize for this issue, and we have corrected it in the revised version.

1. Bruhns P, Jonsson F. Mouse and human FcR effector functions. *Immunol Rev* 2015; **268**(1): 25-51.
2. Talbot J, Hahn P, Kroehling L, Nguyen H, Li D, Littman DR. Feeding-dependent VIP neuron-ILC3 circuit regulates the intestinal barrier. *Nature* 2020; **579**(7800): 575-80.
3. Seillet C, Luong K, Tellier J, et al. The neuropeptide VIP confers anticipatory mucosal immunity by regulating ILC3 activity. *Nat Immunol* 2020; **21**(2): 168-77.
4. Wang Y, Dong W, Zhang Y, Caligiuri MA, Yu J. Dependence of innate lymphoid cell 1 development on NKp46. *PLoS Biol* 2018; **16**(4): e2004867.
5. Zhou L, Zhou W, Joseph AM, et al. Group 3 innate lymphoid cells produce the growth factor HB-EGF to protect the intestine from TNF-mediated inflammation. *Nat Immunol* 2022; **23**(2): 251-61.
6. Zhong C, Cui K, Wilhelm C, et al. Group 3 innate lymphoid cells continuously require the transcription factor GATA-3 after commitment. *Nat Immunol* 2016; **17**(2): 169-78.

7. Di Luccia B, Gilfillan S, Cella M, Colonna M, Huang SC. ILC3s integrate glycolysis and mitochondrial production of reactive oxygen species to fulfill activation demands. *J Exp Med* 2019; **216**(10): 2231-41.
8. Huang J, Lee HY, Zhao X, et al. Interleukin-17D regulates group 3 innate lymphoid cell function through its receptor CD93. *Immunity* 2021; **54**(4): 673-86 e4.
9. Gladiator A, Wangler N, Trautwein-Weidner K, LeibundGut-Landmann S. Cutting edge: IL-17-secreting innate lymphoid cells are essential for host defense against fungal infection. *J Immunol* 2013; **190**(2): 521-5.
10. Huang Y, Guo L, Qiu J, et al. IL-25-responsive, lineage-negative KLRG1(hi) cells are multipotential 'inflammatory' type 2 innate lymphoid cells. *Nat Immunol* 2015; **16**(2): 161-9.
11. Tong Y, Tang J. *Candida albicans* infection and intestinal immunity. *Microbiol Res* 2017; **198**: 27-35.
12. Ost KS, O'Meara TR, Stephens WZ, et al. Adaptive immunity induces mutualism between commensal eukaryotes. *Nature* 2021; **596**(7870): 114-8.
13. Saithong S, Saisorn W, Visitchanakun P, Sae-Khow K, Chiewchengchol D, Leelahavanichkul A. A Synergy Between Endotoxin and (1->3)-Beta-D-Glucan Enhanced Neutrophil Extracellular Traps in *Candida* Administered Dextran Sulfate Solution Induced Colitis in FcGR1B^{-/-} Lupus Mice, an Impact of Intestinal Fungi in Lupus. *J Inflamm Res* 2021; **14**: 2333-52.
14. Bar E, Whitney PG, Moor K, Reis e Sousa C, LeibundGut-Landmann S. IL-17 regulates systemic fungal immunity by controlling the functional competence of NK cells. *Immunity* 2014; **40**(1): 117-27.
15. Klose CS, Artis D. Innate lymphoid cells as regulators of immunity, inflammation and tissue homeostasis. *Nat Immunol* 2016; **17**(7): 765-74.
16. Jarade A, Di Santo JP, Serafini N. Group 3 innate lymphoid cells mediate host defense against attaching and effacing pathogens. *Curr Opin Microbiol* 2021; **63**: 83-91.

REVIEWER COMMENTS

Reviewer #1 (Remarks to the Author):

In their study, Huang et al. explore the potential role of the FcεRIγ chain in ILC3 functions. As noted in the original review these observations provide interesting observations of FcεRIγ+ ILC3s that are novel and of interest. While the authors include essential elementary controls, it is important to note that the main conclusions may require further support from additional experimental approaches. The absence of mechanism associated with the FcεRIγ+ in ILC3s is a matter of concern. While CD16 appears to be involved, the ligand remains unknown. It may be difficult to draw definitive conclusions from this, and further research may be necessary. The authors showed that IgGs affect ILC3 functions in the *C. alb* model but they are unable to bind receptors on the ILC3 surface. What should the reader conclude? Furthermore, the authors have initiated the study of signaling pathways using mutated proteins. However, due to technical difficulties, our understanding is hampered, and it seems risky to base a potential cell signaling in ILC3s on macrophagic cell lines. In addition, the multiplying inflammatory models, without connection (and opposite result) or explanation between them, do not help in understanding the mechanism. Comments to the answers below:

Original Figure1

Question (Q)1

These points have been clarified.

Q2

FcεR1g signaling pathways are well known in macrophages or NK cells, and this analysis does not shed any light on the specific signaling of FcεR1g in ILC3s, particularly on the IL-22 and IL-17A pathways presented here. Based on these results, it seems presumptuous to extrapolate the FcεR1g signaling pathway in ILC3s from different models and cell lines. While the possibility remains, the presented results do not provide conclusive evidence to support the claim. Regarding the fig.7+S8, CD16 depletion (in *Fcgr3*^{-/-} mice) could affect cytokine production in ILC3s but it could also affect the cytokine production by mononuclear phagocytes (ex: den Dunnen et al. Blood 2012; Kozicky et al. JLB 2015) and indirectly the ILC3 biology. Thus, these results don't provide evidence for a functional role of CD16 in ILC3s.

Q3

The results present some contradictions and may benefit from additional investigation. It is suggested that the 'data not shown' (Reviewer Fig. 2) be included in the paper. Exploring the potential effect of immune complexes on ILC3s could be a valuable mechanism to investigate. The text does not provide clarity on whether the authors tested for a possible leakage of Cre recombinase expression in myeloid cells. The presence or absence of FcεR1γ could potentially affect IgG binding and the expression of cytokines, such as IL-23 and IL-1b, which play a crucial role in the maturation and/or activation of ILC3s.

Original Figure2

Q1

Based on the results obtained and the technical problems, it is not reasonably possible to support this conclusion. The authors should move this part to a supplementary figure and revise their conclusion.

Original Figure3

Q1

I don't understand why the authors used different stimulation conditions, but now that the authors have specifically described the protocols in the text, the results seem more logical.

Q2

The authors did not address this question. All the justifications given here are for the *C.albicans* model and not for the *citrobacter* model, it would be preferable to include this part as a supplementary figure to support the results with *C.Alb* and focus the paper on this model for greater clarity. This is particularly important since the entire mechanism is demonstrated in *C. albicans* and not in CR or DSS.

Q3

It's not out of the scope, since these are the results presented by the authors. It seems essential to me that this should at least be discussed.

Q4
Ok

Original Figure4/5

Q1

- Transcriptional analyses show that only CCR6+ ILC3s are affected (S4B), whereas Fig. 5 shows that all ILC3s are affected at the protein level. However, one important piece of data is missing: IL-22 and IL-17A production in ILC3s in mice before infection, it is essential to ensure ILC3 activation in a systemic infection model.

- If CD16 receptors expressed on ILC3s are unable to bind IgG (reviewer fig2), how do the authors explain the effect of mIgG in antifungal immunity? This is not an optimal experimental design to support the non-redundant role of ILC3s and the function of FcεRIγ.

Q2

My question was about Fig.5. The revised Fig.1 and S1 only exhibit the expression of FcR receptors and the different proportions of ILC3s. Because the analysis of RorcCre+Fcr1gf/f PBS is missing in Fig5, it appears that these figures do not provide sufficient transcriptomic analysis to fully understand the effect of the protein on ILC3s before and after infection. The authors did find that FcεRIγ-deficient ILC3s in *C.alb* showed a significant decrease in the expression of infection-induced activation genes. However, I am unclear if the authors observed the same at steady state.

Q3

According to previous findings, while Rag mice were successfully rescued through IgG transfer, it is important to note that the IgGs are unable to bind to the CD16- FcεRIγ complex present on the surface of ILC3s. As a result, their cell intrinsic function in ILC3s cannot be supported.

Minor comments:

- Fig1C: To improve clarity for the reader, it may be beneficial to separate the upper and lower panels, as the lower panel is only present in the RorcCRE+Fcer1g+/+mice
- It would be more appropriate to present the FACS plots (1F) before the bar graphs (1D/1E) and show the frequency of NKp46+ ILC3s in a separate figure.
- S1H- Please indicate "RorcCRE-Fcer1gF/F and RorcCRE+Fcer1gF/F" on the corresponding flow plot
- it would be helpful to include the indicated mice with the corresponding FACS plots for F7A,7E, S8A, S8C:
- S1H: Please specify "cLP" and "siLP" for each line.

Reviewer #2 (Remarks to the Author):

The manuscript has addressed the concerns from the previous review, and is now considered to be a very nice and thorough manuscript with the newly added data. The only remaining request for edits is in Figure 7A and 7E, it is not clear if the different rows within these panels represent different genetic backgrounds. Row labels would help with this issue.

Reviewer #3 (Remarks to the Author):

The revised manuscript is greatly improved. Although the mechanism of how ILC3s in intestine respond to *Candida albicans* that were infected intravenously is still unknown, the data indicate that ILC3s in intestine play a role in the defense against systemic *C. albicans* infection. The authors also provided new data indicate that CD16/FcεRIγ-mediated response in ILC3s is important for antifungal immunity. My concerns have been adequately addressed.

Minor comment:

1. Labeling in Figures: The labeling is missing for Figure 2D, Figure 7A, 7E, S8A, and S8B. For example, I speculate that the top is WT, the middle is *Ncr1*^{-/-}, and the bottom is *Fcgr3*^{-/-}, but the authors should label them clearly.

Response to Reviewer #1

In their study, Huang et al. explore the potential role of the FcεRIγ chain in ILC3 functions. As noted in the original review these observations provide interesting observations of FcεRIγ+ ILC3s that are novel and of interest. While the authors include essential elementary controls, it is important to note that the main conclusions may require further support from additional experimental approaches. The absence of mechanism associated with the FcεRIγ+ in ILC3s is a matter of concern. While CD16 appears to be involved, the ligand remains unknown. It may be difficult to draw definitive conclusions from this, and further research may be necessary. The authors showed that IgGs affect ILC3 functions in the C. alb model but they are unable to bind receptors on the ILC3 surface. What should the reader conclude? Furthermore, the authors have initiated the study of signaling pathways using mutated proteins. However, due to technical difficulties, our understanding is hampered, and it seems risky to base a potential cell signaling in ILC3s on macrophagic cell lines. In addition, the multiplying inflammatory models, without connection (and opposite result) or explanation between them, do not help in understanding the mechanism. Comments to the answers below:

We greatly appreciate the reviewer's valuable suggestion in improving our manuscript.

Original Figure1

Question (Q)1

These points have been clarified.

Q2

FcεRIγ signaling pathways are well known in macrophages or NK cells, and this analysis does not shed any light on the specific signaling of FcεRIγ in ILC3s, particularly on the IL-22 and IL-17A pathways presented here. Based on these results, it seems presumptuous to extrapolate the FcεRIγ signaling pathway in ILC3s from different models and cell lines. While the possibility remains, the presented results do not provide conclusive evidence to support the claim. Regarding the fig.7+S8, CD16 depletion (in Fcgr3-/- mice) could affect cytokine production in ILC3s but it could also affect the cytokine production by mononuclear phagocytes (ex: den Dunnen et al. Blood 2012; Kozicky et al. JLB 2015) and indirectly the ILC3 biology. Thus, these results don't provide evidence for a functional role of CD16 in ILC3s.

We appreciate the reviewer for the comment. To the best of our knowledge, our study is the first to report the role of FcεRIγ in maintaining the expression of phenotypic receptors and

effector function in ILC3s. While FcεR1γ signaling pathway might be conserved in macrophages and ILC, our work demonstrated a non-redundant function of FcεR1γ in ILC3s during infections, advancing our understanding of ILC biology.

Our previous data from IP-MS of RAW cells suggested that FcεR1γ interacted with JAK-STAT and may be involved in activation cascade transduction. In the revised manuscript, we provide new data showing that the deficiency of FcεR1γ in ILC3s significantly impaired the phosphorylation of JAK1 and JAK3 in ILC3s under the infection (**Revised Fig. 5H, 5I**). Taken together with data from the transcriptome analysis and cytokine production of CKO ILC3s (**Fig. 5**), these results provide strong evidence that the JAK-STAT signaling pathway is involved in FcεR1γ-mediated proinflammatory cytokine production in ILC3s. We thank the reviewer for motivating this set of experiments.

Using CKO (*Rorc-cre⁺Fcer1g^{fl/fl}*) mouse model, we have demonstrated a cell-intrinsic function of FcεR1γ in ILC3s. If the decreased cytokine production from ILC3s in *Fcgr3^{-/-}* mice was resulted from the defect of IgG-Fc signaling in mononuclear phagocytes (**Fig. 7A, 7B, S8A and S8B**), as the reviewer has raised concern, cytokine production from ILC3s in the *Rag1^{-/-}* CKO mice would be comparable to that from *Rag1^{-/-}* control mice after replenishment of mouse IgG, since Fc signaling was intact in mononuclear phagocytes. However, we observed impaired cytokine production from ILC3s in the *Rag1^{-/-}* CKO+IgG mice and accelerated mortality compared with *Rag1^{-/-}* control+IgG mice (**Fig. 7D, 7E, 7F, S8C and S8D**). Thus, the results from CKO mice, *Fcgr3^{-/-}* mice and *Rag1^{-/-}* CKO mice together provide evidence for a functional role of CD16 in ILC3s.

Q3

The results present some contradictions and may benefit from additional investigation. It is suggested that the 'data not shown' (Reviewer Fig. 2) be included in the paper. Exploring the potential effect of immune complexes on ILC3s could be a valuable mechanism to investigate. The text does not provide clarity on whether the authors tested for a possible leakage of Cre recombinase expression in myeloid cells. The presence or absence of FcεR1γ could potentially affect IgG binding and the expression of cytokines, such as IL-23 and IL-1b, which play a crucial role in the maturation and/or activation of ILC3s.

As requested by the reviewer, we have included the results from the *in vitro* binding assay in the **Revised Fig. S9**. To address concerns about the potential leakage of Cre recombinase expression in myeloid cells, we have re-analyzed our flow data to check the expression of FcεR1γ in CD45⁺CD3⁻CD19⁻ cells. The specific deletion of FcεR1γ expression in RORγt-expressing ILCs was confirmed in CKO mice compared to control mice, while FcεR1γ expression in RORγt⁻ cells (mainly myeloid cells) was unaffected (**For Reviewer Fig. 1 below**). Furthermore, macrophages in the CKO mice expressed a comparable amount of CD64 (**For Reviewer Fig. 2 below**), the abundance of which is dependent on the presence of FcεR1γ. These results provide evidence against the possibility of Cre leakage to myeloid cells.

For Reviewer Fig. 1

For Reviewer Fig. 2

Original Figure2

Q1

Based on the results obtained and the technical problems, it is not reasonably possible to support this conclusion. The authors should move this part to a supplementary figure and revise their conclusion.

We have successfully verified the significance of aspartic acid at position 29 and leucine at position 39 in ILC3s by utilizing primary ILC3 cells. Our results clearly demonstrate the importance of these amino acids in FcεR1γ for maintaining the abundance of its binding partners across various cell types. Considering the broad relevance of these receptors in immune responses, we believe that our findings will be of significant interest to a wide range of readers. Therefore, we would like to maintain these important findings in the main figure.

Original Figure3

Q1

I don't understand why the authors used different stimulation conditions, but now that the authors have specifically described the protocols in the text, the results seem more logical.

Q2

The authors did not address this question. All the justifications given here are for the C.albicans model and not for the citrobacter model, it would be preferable to include this part as a supplementary figure to support the results with C.Alb and focus the paper on this model for greater clarity. This is particularly important since the entire mechanism is demonstrated in C. albicans and not in CR or DSS.

We apologize for any confusion caused. In our study, we utilized the *C. rodentium* infection model, which is a well-established model used to investigate ILC3 biology (Ref #12). Upon infection, ILC3 release IL-22, which activates the IL-22R-JAK1-STAT signal transduction pathway in intestinal epithelial cells. This activation leads to the release of antimicrobial peptides and mucus, aiding in the clearance of the pathogen. Since the mechanisms underlying this immunological cascade have been extensively studied, we did not further investigate the *C. rodentium* model in our current study. Instead, we focused on the *C. albicans* model. In both the *C. rodentium* and *C. albicans* models, we observed impaired cytokine production in ILC3s

of CKO mice. Therefore, we propose that the CD16-FcεR1γ signaling pathway influences the inflammatory cytokine production of ILC3s in both of these models.

Q3

It's not out of the scope, since these are the results presented by the authors. It seems essential to me that this should at least be discussed.

We thank the reviewer for the suggestion. Our data demonstrated that CD16 functions as a co-receptor of FcεR1γ on ILC3s, contributing to anti-infection immunity. Considering that the binding with CD16 with IgG is influenced by its format, such as being in an immune complex, aggregated or opsonized by other components, we propose that the potential difference in antibody production and their engagement pattern with CD16-FcεR1γ complex on ILC3s under different conditions, including infection, DSS treatment and homeostasis, may explain why CD16 deficiency resulting from FcεR1γ knockout leads to a defect in cytokine production only during infection. We have discussed this part in the revised **Discussion** (p.14).

Q4

Ok

Original Figure4/5

Q1

- Transcriptional analyses show that only CCR6+ ILC3s are affected (S4B), whereas Fig. 5 shows that all ILC3s are affected at the protein level. However, one important piece of data is missing: IL-22 and IL-17A production in ILC3s in mice before infection, it is essential to ensure ILC3 activation in a systemic infection model.

We appreciate the reviewer's comment. As requested, we have included new data showing that the loss of FcεR1γ did not influence the production of IL-17A and IL-22 in intestinal ILC3s at the steady state (**Revised Fig. S11**). Thus the reduction of cytokine production in FcεR1γ-deficient ILC3s is a specific phenotype observed only during infection (**Fig. 5**).

- If CD16 receptors expressed on ILC3s are unable to bind IgG (reviewer fig2), how do the authors explain the effect of mIgG in antifungal immunity? This is not an optimal experimental design to support the non-redundant role of ILC3s and the function of FcεRIγ.

Our *in vivo* studies demonstrated that IgG-CD16 pathway promotes cytokine production and anti-fungal immunity in ILC3s. However, the *in vitro* binding assay (**Revised Fig. S9**) did not show direct binding of IgG on ILC3s. This discrepancy may be attributed to the low-to-medium affinity Fc receptors, including CD16, can only bind IgG when it is present in an immune complex, aggregated, or opsonized by other component (Ref #15). On the other hand, high-affinity Fc receptors, such as CD64, can bind free/monomeric IgG. The potential difference in antibody production and their engagement pattern with CD16-FcεRIγ complex on ILC3s under different conditions, including infection, DSS treatment and homeostasis, may explain why CD16 deficiency resulting from FcεRIγ knockout leads to a defect in cytokine production only during infection.

Q2

My question was about Fig.5. The revised Fig.1 and S1 only exhibit the expression of FcR receptors and the different proportions of ILC3s. Because the analysis of RorcCre+Fcr1gf/f PBS is missing in Fig5, it appears that these figures do not provide sufficient transcriptomic analysis to fully understand the effect of the protein on ILC3s before and after infection. The authors did find that FcεRIγ-deficient ILC3s in C.alb showed a significant decrease in the expression of infection-induced activation genes. However, it is unclear if the authors observed the same at steady state.

In **Fig. S1L**, we have showed that there were not many DEGs between CKO ILC3s and control ILC3s at the steady state in our scRNA-seq dataset. Additionally, we performed bulk RNA-seq analysis on sorted total ILC3s from siLPs and cLPs at the steady state. In this analysis, we also did not observe a significant difference in the expression of those infection-induced activation genes between control and FcεRIγ-deficient ILC3s (**For Reviewer Fig. 3 below**). To avoid repetition of the scRNA-seq data with the bulk RNA-seq data, we did not include these results in the manuscript.

For Reviewer Fig. 3

FC>1.2, p.adj<0.05

Q3

According to previous findings, while Rag mice were successfully rescued through IgG transfer, it is important to note that the IgGs are unable to bind to the CD16- FcεRIγ complex present on the surface of ILC3s. As a result, their cell intrinsic function in ILC3s cannot be supported.

We kindly direct the reviewer to the detailed response to this comment in our response to the Q1 above, and in the revised **Discussion** (p. 14 and 15).

Minor comments:

- Fig1C: To improve clarity for the reader, it may be beneficial to separate the upper and lower panels, as the lower panel is only present in the RorcCRE+Fcer1g+/+mice

To improve clarity, we have labeled the genotype for the lower panel in the **Revised Fig. 1C**.

- It would be more appropriate to present the FACS plots (1F) before the bar graphs (1D/1E) and show the frequency of NKp46+ ILC3s in a separate figure.

We appreciate the reviewer's suggestion. CCR6⁺ and CCR6⁻ ILC3s represent two distinct ILC3 subsets. In **Fig. 1D** and **1E**, we showed that the loss of FcεRIγ did not affect the frequencies of these two subsets. We then further analyzed the expression of NKp46, which is a phenotypic protein that is only expressed on CCR6⁻ ILC3 subset (**Fig. 1F**). So we believe that this presentation is more logical and appropriate.

- *S1H*- Please indicate “RorcCRE-Fcer1gF/F and RorcCRE+Fcer1gF/F “on the corresponding flacs plot

- it would be helpful to include the indicated mice with the corresponding FACS plots for F7A,7E, S8A, S8C:

-*S1H*: Please specify “cLP” and “siLP” for each line.

We have revised all these panels (**Revised Fig. S1H, 7A, 7E, S8A, S8C**) accordingly.

Reviewer #2 (Remarks to the Author):

The manuscript has addressed the concerns from the previous review, and is now considered to be a very nice and thorough manuscript with the newly added data. The only remaining request for edits is in Figure 7A and 7E, it is not clear if the different rows within these panels represent different genetic backgrounds. Row labels would help with this issue.

We thank the reviewer for the appreciation of our revised work. We have revised **Fig. 7A** and **7E** accordingly.

Reviewer #3 (Remarks to the Author):

The revised manuscript is greatly improved. Although the mechanism of how ILC3s in intestine respond to Candida albicans that were infected intravenously is still unknown, the data indicate that ILC3s in intestine play a role in the defense against systemic C. albicans infection.

The authors also provided new data indicate that CD16/FcεR1γ-mediated response in ILC3s is important for antifungal immunity. My concerns have been adequately addressed.

We thank the reviewer for the appreciation of our revised work.

Minor comment:

1. Labeling in Figures: The labeling is missing for Figure 2D, Figure 7A, 7E, S8A, and S8B. For example, I speculate that the top is WT, the middle is Ncr1^{-/-}, and the bottom is Fcgr3^{-/-}, but the authors should label them clearly.

We have revised these panels accordingly.

REVIEWERS' COMMENTS

Reviewer #1 (Remarks to the Author):

I would like to thank the authors for their work and the constructive discussion that took place during the revision process. While the mechanism may remain incomplete, I have no further comments at this stage.

Response to Reviewer #1

I would like to thank the authors for their work and the constructive discussion that took place during the revision process. While the mechanism may remain incomplete, I have no further comments at this stage.

We thank the reviewer for the appreciation of our revised work.